# Ferromagnetism and giant magnetoresistance in zinc-blende FeAs monolayers embedded in semiconductor structures

Le Duc Anh [1,2,3✉], Taiki Hayakawa[1], Yuji Nakagawa [4], Hikari Shinya [5,6,7], Tetsuya Fukushima[7,8,9], Masaki Kobayashi[1,9], Hiroshi Katayama-Yoshida[9], Yoshihiro Iwasa [4,10] & Masaaki Tanaka [1,9✉]

Material structures containing tetrahedral FeAs bonds, depending on their density and geometrical distribution, can host several competing quantum ground states ranging from superconductivity to ferromagnetism. Here we examine structures of quasi two-dimensional (2D) layers of tetrahedral Fe-As bonds embedded with a regular interval in a semiconductor InAs matrix, which resembles the crystal structure of Fe-based superconductors. Contrary to the case of Fe-based pnictides, these FeAs/InAs superlattices (SLs) exhibit ferromagnetism, whose Curie temperature ($T_C$) increases rapidly with decreasing the InAs interval thickness $t_{InAs}$ ($T_C \propto t_{InAs}^{-3}$), and an extremely large magnetoresistance up to 500% that is tunable by a gate voltage. Our first principles calculations reveal the important role of disordered positions of Fe atoms in the establishment of ferromagnetism in these quasi-2D FeAs-based SLs. These unique features mark the FeAs/InAs SLs as promising structures for spintronic applications.

[1] Dept. of Electrical Engineering and Information Systems, The University of Tokyo, Tokyo, Japan. [2] Institute of Engineering Innovation, The University of Tokyo, Tokyo, Japan. [3] PRESTO, Japan Science and Technology Agency, Tokyo, Japan. [4] QPEC & Dept. of Applied Physics, The University of Tokyo, Tokyo, Japan. [5] Research Institute of Electrical Communication, Tohoku University, Miyagi, Japan. [6] Center for Spintronics Research Network (CSRN), Tohoku University, Miyagi, Japan. [7] Center for Spintronics Research Network (CSRN), Osaka University, Osaka, Japan. [8] Institute for Solid State Physics, The University of Tokyo, Chiba, Japan. [9] Center for Spintronics Research Network (CSRN), The University of Tokyo, Tokyo, Japan. [10] RIKEN Center for Emergent Matter Science (CEMS), Saitama, Japan. ✉email: anh@cryst.t.u-tokyo.ac.jp; masaaki@ee.t.u-tokyo.ac.jp

Tetrahedral FeAs-based materials have attracted much attention in recent years since the discovery of Fe-based high-temperature superconductors (Fe-based pnictides)[1,2] and later, Fe-based high-$T_C$ ferromagnetic semiconductors (FMSs)[3–11]. In Fe-based superconductors where Fe-As bonds are confined in 2D monolayers (MLs), the magnetic ground state of the Fe spins is antiferromagnetic by super-exchange interaction, and superconductivity appears upon doping carriers or applying pressure. On the other hand, in Fe-based FMSs such as (In,Fe)As where Fe-As bonds are distributed randomly in a three-dimensional (3D) InAs semiconductor matrix, the Fe spins couple ferromagnetically via interactions with electron carriers[3,6,7]. These results strongly suggest that the density and geometrical distribution of the Fe-As bonds are crucial in determining the transport and magnetic properties of their nanostructures. To understand the underlying physics behind these fascinating structures, it is necessary to systematically study their transport and magnetic properties when varying the distribution of Fe-As bonds. In particular, it is required and important to investigate the nature of the magnetic ground state in the 3D-2D crossover limit when the Fe-As bonds are confined in 2D ultrathin layers embedded in the InAs matrix at a regular distance, which resembles the case of both the Fe-based pnictides and (In, Fe)As FMS (Fig. 1a).

A pioneering theoretical work by Griffin and Spaldin[12] suggested that in superlattice (SL) structures of FeAs tetrahedral layers embedded in a zinc-blende semiconductor matrix, the antiferromagnetic phase should be the magnetic ground state as in the Fe-based superconductors. The work also revealed a half-metallic band structure in the hypothetical ferromagnetic (FM) phase of zinc blende FeAs. On the other hand, in $(In_{1−x}Fe_x)As$, where Fe atoms ($x = 1 – 10\%$) are randomly distributed and partially replace the In sites and thus the local Fe density is much less than that of the FeAs ML-based SLs, previous experimental works revealed electron-induced ferromagnetism with Curie temperature ($T_C$) as high as 120 K[13,14]. This result was striking since the $s$-$d$ exchange interaction was hitherto believed to be extremely weak[15,16]. In fact, (In,Fe)As is the first $n$-type electron-induced III-V FMS, which provides an important missing piece for spin device applications based on semiconductors. Various unique and essential features have been realized in this material, such as quantum size effects[5,6], large $s$–$d$ exchange interaction energy[4,6], large spontaneous spin splitting in the conduction band[7], and proximity-induced superconductivity over a very long distance (~1 μm)[13,14]. A notable feature in (In,Fe)As and other Fe-based FMSs is the rapid increase of $T_C$ over room temperature with increasing the Fe density[11]. Indeed, first principles calculations suggested that the FM interactions between the Fe atoms in InAs can be strongly enhanced if the distance between the Fe atoms can be reduced to that between the next-nearest neighbor sites, via super-exchange interactions[17]. Furthermore, enhancement of ferromagnetism has also been reported in similar magnetic digital alloys of Mn-doped FMSs[18–20]. These recent experimental and theoretical findings thus suggested that in the FeAs-based SLs, a FM ground state is favorable. These contradicting theoretical predictions on the dependence of the magnetic properties of Fe doped InAs on the geometrical distribution of the Fe-As bonds thus strongly call for experimental verifications.

In this work, we study the epitaxial growth and properties of SL structures of FeAs quasi-2D ultrathin layers embedded in an InAs semiconductor matrix at a regular distance ($t_{InAs}$). Contrary to the first principle calculation of ref. 12 and the case of Fe-based pnictides, we find that these FeAs/InAs SLs exhibit strong ferromagnetism whose $T_C$ increases rapidly with decreasing the InAs interlayer thickness $t_{InAs}$ ($T_C \propto t_{InAs}^{-3}$), with a very high magnetic moment per Fe atom (4.7–4.9 $\mu_B$, where $\mu_B$ is Bohr magneton). We also observe in these SL structures an extremely large magnetoresistance (MR) up to 500%, which is tunable by a gate voltage. Our microstructure characterizations and first principles calculations reveal that disordered positions of Fe atoms play a key role in establishing the FM ground state. These unique features indicate that these FeAs/InAs SLs are promising for spintronic applications.

## Results

**Growth and crystal structure of FeAs/InAs SL structures.** Growth of FeAs/InAs SLs with zinc-blende crystal structure is highly challenging due to the low solubility of Fe in III–V semiconductors, which easily causes atomic segregation and

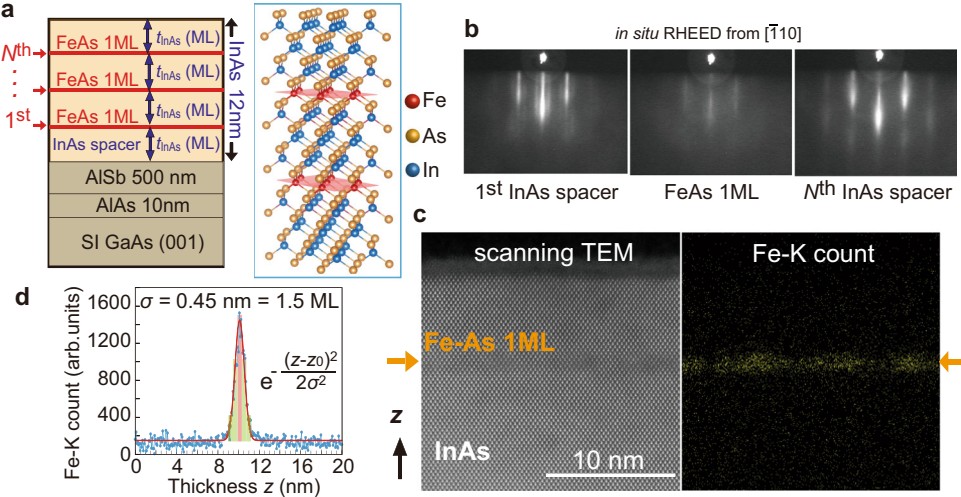

**Fig. 1 Crystal growth of FeAs/InAs SL structures. a** Schematic sample structure of the FeAs/InAs SL grown on a GaAs (001) substrate. There are $N$ FeAs layers (red planes) embedded in an InAs matrix with a regular distance of $t_{InAs}$. The total thickness of the FeAs/InAs structures is 41–47 MLs (= 12.4–14 nm). **b** In situ RHEED patterns along the [1̄10] direction during the growth of the FeAs/InAs SL in sample A4. **c** High-resolution scanning TEM of a sample with 1 ML FeAs embedded in InAs (left panel) and the corresponding mapping of Fe atoms by EDX (right panel). **d** Fe distribution along the growth direction mapped with EDX, which follows a normal distribution with a standard deviation of 0.45 nm (1.5 ML of InAs).

**Table 1 Structure parameters ($N$, $t_{InAs}$) and Curie temperature $T_C$ of sample A0 – A4.**

| Sample | FeAs layer number $N$ | FeAs interlayer distance $t_{InAs}$ (ML) | SL total thickness (nm) | $T_C$ (K) |
|---|---|---|---|---|
| A0 (ref) | NA | NA | 12.0 | 26 |
| A1 | 1 | 20 | 12.4 | 5 |
| A2 | 3 | 10 | 13.0 | 10 |
| A3 | 5 | 7 | 14.2 | 25 |
| A4 | 7 | 5 | 14.2 | 80 |

All samples (except sample A0) have a FeAs/InAs superlattice (SL) structure with a total thickness of 41–47 monolayers (MLs), corresponding to 12.4–14.2 nm.

phase separation. Indeed, there is no report on the zinc-blende FeAs bulk structure to date although it is theoretically predicted to host a half-metallic band structure[12]. In this work, we grew FeAs/InAs SL structures on InAs (001) or semi-insulating GaAs (001) substrates by employing a special technique of low-temperature molecular-beam epitaxy (LT-MBE). In the growth on GaAs substrates, as illustrated in Fig. 1a, we grow a 500-nm-thick AlSb layer at a growth temperature $T_S = 470\,°C$ on a 10-nm-thick AlAs/GaAs (001) substrate prior to the growth of the SL structures to obtain a smooth and lattice-matched buffered layer. Then $T_S$ is decreased to 220 °C and we grow the FeAs/InAs SL structure, consisting of $N$ periods of InAs ($t_{InAs}$ MLs)/FeAs (nominally 1 ML), and finally an InAs cap layer ($t_{InAs}$ MLs). The InAs and FeAs layers are grown at growth rates of 500 nm/h and 250 nm/h, respectively, for which we calibrated the Fe flux to be exactly a half of the In flux (see "Methods"). We grew a series of samples, numbered from A1 to A4, with the thickness parameters ($N$, $t_{InAs}$) = (1, 20), (3,10), (5, 7), (7, 5), respectively. Here, the total thicknesses of the FeAs/InAs SLs in all the samples (A1–A4) are nearly fixed at 41–47 MLs, which are equal to 12.4–14 nm. Therefore the distance between the FeAs layers, $t_{InAs}$, is inversely proportional to $N$. For reference, we also grow sample A0 consisting of a 12-nm-thick structure of (In,Fe)As (6% Fe, 7 nm)/ InAs (5 nm) on top of an AlSb buffer layer. The structure and $T_C$ of all the samples are summarized in Table 1. In Fig. 1b, we show the in situ reflection high energy electron diffraction (RHEED) patterns of the SL structure in sample A4 ($N = 7$). The RHEED patterns are very bright and streaky during the growth of the InAs layers, darken but remain the zinc-blende pattern during the growth of the FeAs layers, then well recovered during the growth of the next InAs layer. This indicates that the SL structures in all the samples are grown in a good 2D growth mode maintaining the zinc-blende crystal structure, and there is no sign of second-phase precipitation.

We examine the crystal structure and Fe distribution using scanning transmission electron microscopy (STEM) and energy-dispersive X-ray spectroscopy (EDX) mapping. Figure 1c shows the results of a sample consisting of a FeAs (nominally 1 ML) sandwiched between InAs layers, grown on an InAs (001) substrate using the same growth procedure and conditions explained above. Only the zinc-blende crystal structure of the host InAs is observed from the STEM lattice image (left panel of Fig. 1c). The STEM contrast is directly related to the atomic number, so that one can see a darker horizontal line with a line width of roughly 3 MLs corresponding to the FeAs layer. EDX mapping results of the Fe atoms also confirm the Fe distribution in a quasi-ML located at 10 nm from the surface (right panel of Fig. 1c). Along the growth direction $z$, the Fe concentration follows a normal distribution with a standard deviation $\sigma = 0.45$ nm, corresponding to 1.5 ML using the lattice constant of InAs (Fig. 3d). We note that the spatial resolution of EDX is 3–4 MLs, which is larger than $\sigma$. Therefore, we can conclude that ultrathin tetrahedral FeAs layers are successfully grown in a zinc-blende

InAs matrix, and the Fe atoms are mainly confined within a thickness of 2–3 MLs.

**Magnetic properties of the FeAs/InAs SL structures.** We examine the magnetic properties of sample A0–A4 using magnetic circular dichroism (MCD) and superconducting quantum interference device (SQUID) magnetometry. Figure 2a shows the MCD spectra of these samples measured at 5 K and under a magnetic field **H** of 1 T applied perpendicular to the film plane. The MCD spectra of the samples A1–A4 exhibit very similar shapes with strongly enhanced peaks close to the optical transition energies at the critical points of the zinc-blende InAs band structure, $E_1$ (2.61 eV), $E_1 + \Delta_1$ (2.88 eV), $E_0'$ (4.39 eV), and $E_2$ (4.74 eV)[3]. These features are similar to the MCD spectrum of the $(In_{0.94},Fe_{0.06})As$ reference sample (A0) with the same thickness (12 nm). This result indicates that even when the Fe atoms are highly concentrated in the 2D ultrathin layers, the band structure of the SLs maintains the basic properties of the host InAs, such as the band gap and most of the band dispersions. This finding is consistent with their single-phase zinc-blende crystal structure revealed by STEM. Moreover, compared with the MCD spectrum of the $(In_{0.94},Fe_{0.06})As$ reference sample, the MCD intensity of the FeAs/InAs SL in sample A2, which has almost the same average Fe concentration (which is 6.9% = 3 MLs / 43 MLs), is threefold stronger. This clearly indicates enhancement of the magnetization and the spin splitting energy in the band structure of the FeAs/ InAs SLs, probably because the Fe atoms are closely distributed in the FeAs planes. The MCD intensity in sample A2, A3, A4 stays almost constant, which may be due to a short penetration depth of visible light in these structures (See Supplementary Note 4).

We estimate $T_C$ of sample A1–A4 by the Arrott plots[21] of the MCD–$H$ curves at different temperatures (Fig. 2b), whose results are summarized in Fig. 2c. All the samples exhibit ferromagnetism, contradicting the prediction of ref. 12. Interestingly, $T_C$ increases rapidly as the distance $t_{InAs}$ between the FeAs MLs decreases, which can be approximately expressed as $T_C \propto t_{InAs}^{-3}$. This relation indicates that the interlayer magnetic interaction between the FeAs layers, which is likely mediated by electron carriers, plays an important role in inducing the ferromagnetism. The inverse third power dependence of $T_C$ on $t_{InAs}$ can be understood qualitatively as follows: If we regard the total magnetic moment in each FeAs ML as a macroscopic spin **S**, and define $J_{Sij}$ as the exchange interaction energy between the magnetic moments **S** of the $i$th and $j$th FeAs MLs separated by a distance of $|i–j| t_{InAs}$, $J_{Sij}$ is proportional to $\sim|i–j|^{-2}t_{InAs}^{-2}$ as is well-known in magnetic multilayer systems with RKKY-like interlayer interaction[22]. Using the Heisenberg model, we have the following relation:

$$T_C \propto \frac{1}{2}\sum_{i,j=1\,(i\neq j)}^{N} J_{Sij}S^2 = \frac{S^2}{2t_{InAs}^2}\sum_{i,j=1\,(i\neq j)}^{N}\frac{1}{(i-j)^2} = \frac{S^2}{2t_{InAs}^2}\sum_{i=1}^{N}\frac{N-i}{i^2}$$

(1)

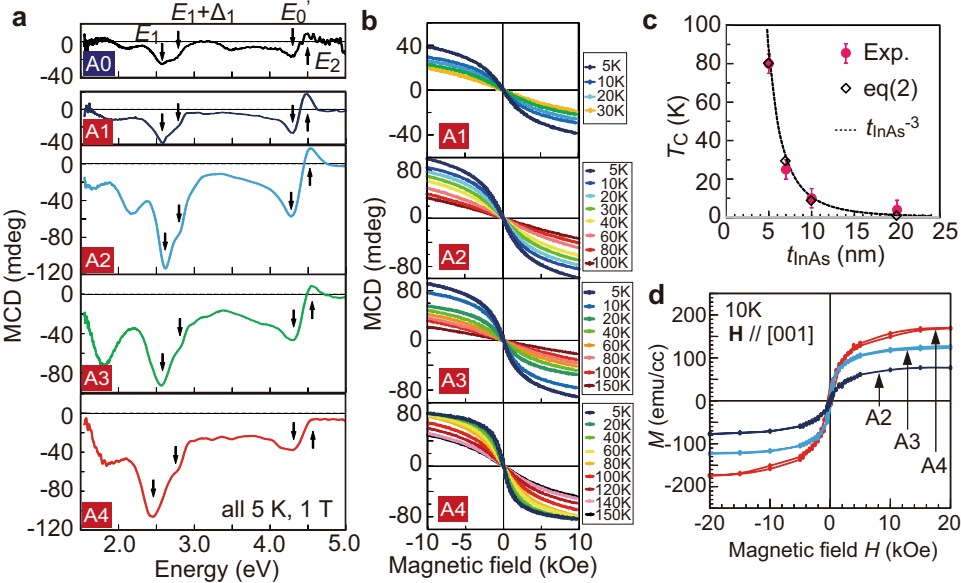

**Fig. 2 Magnetic properties of the FeAs/InAs SLs. a** MCD spectra of sample A1–A4, compared with that of a reference sample (A0) of 12 nm-thick ($In_{0.94}$, $Fe_{0.06}$)As. All spectra were measured at 5 K and 1 T. **b** Magnetic field dependence of MCD intensity (MCD–H curves) measured at $E_1$ in sample A1–A4 at various temperatures. **c** Curie temperature ($T_C$) as a function of the distance $t_{InAs}$ between the FeAs MLs (pink circles). The error bars of 5 K are also plotted, which correspond to the minimum temperature step in our MCD–H measurements. The relationship can be fitted well by the $T_C \propto t_{InAs}^{-3}$ curve (dotted curve). The $T_C$ values calculated using Eq. (2) with $A = 250$ (open diamonds) also well reproduce the experimental results. **d** Magnetic field dependence of magnetization in sample A2–A4, measured by SQUID magnetometry at 10 K, under a magnetic field perpendicular to the film plane.

We deduce from Eq. (1) that

$$T_C = A\left(\frac{N}{t_{InAs}^2}\sum_{i=1}^{N}\frac{1}{i^2} - \frac{1}{t_{InAs}^2}\sum_{i=1}^{N}\frac{1}{i}\right) \qquad (2)$$

where $A$ is a proportional constant. In Eq. (2), the first term is roughly 1.5, 2.5, 3, and 4 times larger than the second term for $N = 1, 3, 5, 7$, respectively. Therefore, considering the relationship $N \propto t_{InAs}^{-1}$, if we express $T_C$ as $t_{InAs}^{-\gamma}$, the exponent $\gamma$ is a value close to 3. As can be seen in Fig. 2c, the $T_C$ values calculated using Eq. (2) with $A = 250$ (open diamonds) well reproduce the experimental results (pink circles). An important implication of this $T_C \propto t_{InAs}^{-3}$ relationship is that we would obtain a room-temperature ferromagnetism in the SLs with $t_{InAs} < 3$ MLs, which might be achievable by optimizing the growth conditions.

Another important feature of the FeAs/InAs SL structures is that they possess a large magnetic moment per Fe atom of 4.7–4.9 $\mu_B$. As shown in Fig. 2d, the saturated magnetization in sample A2–A4, measured at 10 K under a magnetic field perpendicular to the film plane by SQUID magnetometry, increases linearly with the number $N$ of the FeAs layers. These impressive results might be induced by the 2D distribution of the Fe atoms, which are all neighboring to InAs at both the top and bottom interfaces. This lowers the symmetry around the Fe atoms, which can strongly enhance their orbital moment component to as high as that of an isolated Fe atom (2 $\mu_B$/Fe)[23]. This high value of magnetic moment also excludes the possibility of Fe clusters, which usually have a magnetic moment of only 2.2 $\mu_B$/Fe atom (see Supplementary Note 1). This is a huge improvement from the case of randomly Fe-doped (In,Fe)As, in which the magnetic moment per Fe atom is only 1.8 $\mu_B$ corresponding to a missing of 64% of the Fe magnetic moment[3]. This finding agrees well with the prediction of the first principles calculation that the FM interactions between the Fe atoms in InAs can be strongly enhanced if the distance between the Fe atoms is reduced[17].

**Magnetoresistances (MR) in the FeAs/InAs SL structures.** The transport properties of the FeAs/InAs SL structures, examined in patterned Hall bars of size $50 \times 200$ $\mu m^2$, also depend significantly on $t_{InAs}$. As shown in Fig. 3a, the temperature dependence of the resistivity changes from metallic behavior in samples A1 and A2 to insulating behavior in samples A3 and A4. Particularly at low temperatures, the resistivity drastically increases by 5 orders of magnitude when decreasing $t_{InAs}$ from 20 MLs (sample A1) to 5 MLs (sample A4). However, the resistivity decreases significantly when we applied an external magnetic field **H** perpendicular to the film plane. The decrease of resistivity by applying **H** is particularly strong in the samples with high resistivity. Figure 3b shows MR, where the MR ratio is defined as $MR(H) = [R(0) - R(H)]/R(H) \times 100\%$, in sample A4, measured with **H** perpendicular to the film plane at various temperatures. Open and closed symbols in the MR curve at each temperature correspond to the MR ratios under a left-to-right and right-to-left magnetic field sweeping direction, respectively. Figure 3c shows the temperature dependence of MR. Extremely large MR ratios (maximum ~500% at 2 K and 10 kOe) with clear hysteresis were observed. With increasing temperature, the MR ratio decreases remarkably in correlation with the decrease of the resistivity, and shrinks to below 1% at the temperatures above $T_C$ (80 K). Figure 3d shows the MR curves in the four SL samples A1–A4, measured at 4 K with **H** up to 1 T. One can see that the MR ratio increases rapidly from $1 \rightarrow 27 \rightarrow 82 \rightarrow 224\%$ when one goes from sample A1 to A4, in accordance with the decrease of $t_{InAs}$ and the increase of the resistivity.

The strong correlation between the resistivity and the MR ratio clearly implies that spin-dependent scattering of conduction carriers, which are electrons as indicated by Hall measurements, is the dominant scattering mechanism in these FeAs/InAs SL samples. We note that the situation here is very similar to the well-known case of multilayer structures composed of ferromagnetic (FM)/nonmagnetic (NM) metallic layers, in which giant

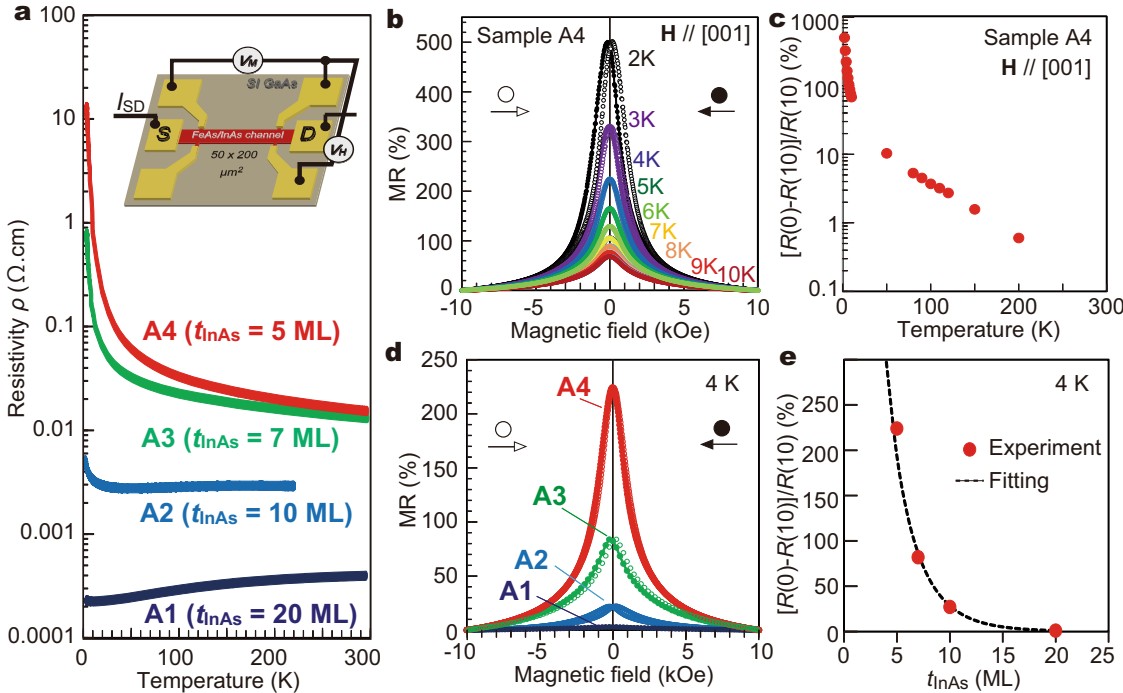

**Fig. 3 Magnetotransport properties of the FeAs/InAs SL structures. a** Temperature dependence of the resistivity in sample A1–A4. The resistivity was measured by the four-terminal method in Hall bars with size of $50 \times 200$ μm$^2$ (inset). **b** Magnetoresistance (MR) curves measured in sample A4 at various temperatures $T = 2$–10 K, when applying a magnetic field perpendicular to the film plane. The MR ratio is defined as MR $(H) = [R(0) - R(H)]/R(H) \times 100\%$. Open and closed symbols in the MR curve at each temperature correspond to the MR ratios under a left-to-right and right-to-left magnetic field sweeping direction, respectively. **c** MR ratio at $H = 10$ kOe as a function of temperature in sample A4. **d** MR curves measured in sample A1–A4 at 4 K when applying a magnetic field perpendicular to the film plane. Open and closed symbols correspond to opposite sweeping directions of magnetic field, as described in (**b**). **e** MR ratio at $H = 10$ kOe as a function of the distance $t_{InAs}$ between the FeAs layers. The relationship can be fitted well by Eq. (4) (dotted curve).

magnetoresistance (GMR) is usually observed. Here, the FeAs layers and InAs spacers play the roles of FM and NM layers, respectively. We thus used a standard two-carrier conduction model usually used for GMR to analyse the large MR observed in our SL samples. The charge conduction was then divided into up-spin and down-spin channels, with the resistivity of $\rho_\uparrow$ and $\rho_\downarrow$, respectively. Using $\rho_\uparrow$ and $\rho_\downarrow$, we can rewrite[24]

$$\text{MR} = \frac{\left(\rho_\downarrow - \rho_\uparrow\right)^2}{4\rho_\downarrow \rho_\uparrow} = \frac{(\alpha - 1)^2}{4\alpha} \quad (3)$$

with $\alpha = \rho_\downarrow/\rho_\uparrow$ is the spin asymmetry parameter. Due to the spin-dependent scattering at the interfaces of the FeAs layers, $\alpha$ largely deviates from 1. Using Eq. (3) for the MR of sample A4 (~500% at 2 K), we have an extremely large $\alpha = 22$. There are two possible reasons for this giant value of $\alpha$: First, it may originate from the large spin-splitting in the conduction band of the FeAs/InAs SLs as revealed by the MCD results (Fig. 2). It is worth noting that for the case of (In,Fe)As we have previously observed the half-metallic conduction band structure[7]. Second, it may originate from high spin-polarization in the DOS of the zinc-blende FeAs layers, because a half-metallic DOS was predicted for bulk FeAs by the first principles calculations[12].

Based on the theoretical framework of GMR[24], we quantitatively explain the $t_{InAs}$-dependence of MR shown in Fig. 3e. In FM/NM multilayers, where GMR appears, the dependence of the MR ratio on the NM layer thickness ($t_{InAs}$) is given by

$$\text{MR} = MR_0 \frac{\exp\left(-\frac{t_{InAs}}{l_{InAs}}\right)}{\left(1 + \frac{t_{InAs}}{d_0}\right)} \quad (4)$$

Here $MR_0$ is the MR magnitude constant, $l_{InAs}$ is the electron mean free path in the InAs spacer, $d_0$ is an effective length representing the shunting of the current in the FeAs layers. As shown in Fig. 3e, Eq. (4) provides a very good fitting to the exponential $t_{InAs}$-dependence of MR with $MR_0 = 2700\%$, $l_{InAs} = 1.1$ nm (~4 MLs of InAs) and $d_0 = 0.6$ nm. The short mean free path $l_{InAs}$ of 4 MLs is nearly equal to the thickness of the InAs spacer in sample A4 ($t_{InAs} = 5$ MLs). This finding is consistent with the fact that the spin-dependent scattering occurs mainly at the FeAs/InAs interfaces. On the other hand, the value $d_0 = 0.6$ nm is close to the broadening ($\sigma = 0.45$ nm) of the FeAs layers observed by EDX and STEM (Fig. 1). These good agreements strongly suggest that the extremely large MRs in our SL samples can be well understood by the GMR model in semiconductor-based magnetic multilayer systems. These findings also suggest that even higher MR ratios may be obtained in the SLs by optimizing the structure parameter, particularly by using smaller $t_{InAs}$.

**Electrical control of GMR in the FeAs/InAs SL structures.** One major advantage of our semiconductor-based magnetic multilayer structures over metallic ones is the ability to control the MR properties using a gate voltage $V_G$. As shown in Fig. 4a, we form an electric double layer field-effect transistor (EDLT) structure using the Hall bar of sample A2 ($N = 3$, $t_{InAs} = 10$ MLs) (see "Methods").

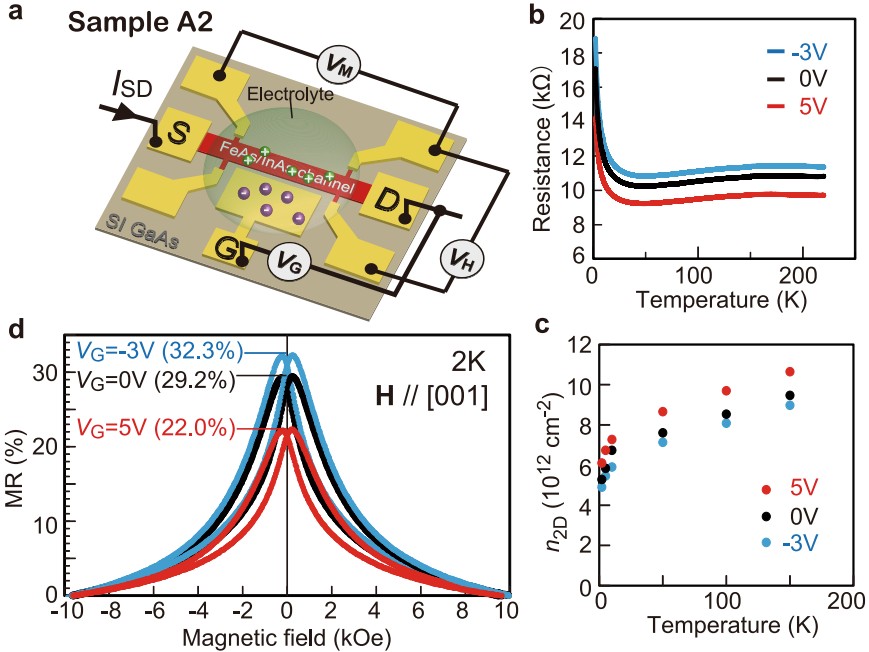

**Fig. 4 Gate voltage control of the magnetoresistance (MR) in FeAs/InAs SLs. a** Electrical double-layer transistor structure formed on sample A2 (see "Method"). **b** Resistance and **c** sheet electron density ($n_{2D}$) in the FeAs/InAs SL structure at various temperatures under different gate voltages $V_G = -3, 0, 5$ V. **d** MR curves measured in the FeAs/InAs SL at 2 K with different gate voltages $V_G = -3, 0, 5$ V, under a magnetic field applied perpendicular to the film plane. These MR curves show clear hysteresis due to the ferromagnetism in sample A2. The values of MR ratio are given in the parentheses after the gate voltages.

As shown in Fig. 4b and c, the resistance and sheet electron concentration $n_{2D}$ in sample A2 are systematically controlled by applying $V_G$. The change in $n_{2D}$ is small, only 15.5% at $V_G = 5$ V and $-7.5\%$ at $V_G = -3$ V. However, as shown in Fig. 4d, significant changes in the MR curves are obtained: The MR ratio at $H = 10$ kOe decreases from 29.2% at $V_G = 0$ V to 22% at $V_G = 5$ V, and increases to 32.3% at $V_G = -3$ V. Again, an increase (decrease) of the MR ratio in accordance with an increase (decrease) of the resistivity is observed. With increasing the electron concentration, the spin-dependent scattering of electron carriers at the FeAs/InAs interface is suppressed due to an enhanced screening effect, which leads to the corresponding decrease in the MR ratio.

## Discussions
The remaining main question is how the FM ground state is induced in the tetrahedral FeAs/InAs SLs of the zinc-blende crystal structure, which contradicts the prediction by the first principles calculations[12]. Indeed, by performing first principles calculations on a FeAs/InAs SL structure with $t_{InAs} = 5$ MLs (see "Methods"), we deduce a similar conclusion that the magnetic exchange interactions between the substitutional Fe atoms (Fe$_\delta$) in a 1 ML FeAs of the SL structure are antiferromagnetic with an exchange energy $J_{ij} = -23$ meV. This result thus suggests that the ground state of the SL system should be antiferromagnetic if all the Fe atoms are ideally distributed within the 1 ML thickness of the delta-doping zinc-blende FeAs layers (Fe$_\delta$, the red atoms in Fig. 5a, b). However, we surprisingly found that if there is a small amount of disorder where Fe atoms are located in the octahedral interstitial sites (Fe$_i$, purple atoms in Fig. 5a) and/or in the As sites (Fe$_{As}$, green atoms in Fig. 5b) which are very close to the 1ML-thick FeAs layer (red planes in Fig. 5a and b), a remarkably stable FM ground state can be established. As shown in Fig. 5c, d, the magnetic interactions between the Fe$_i$ or Fe$_{As}$ and the Fe$_\delta$ is FM with a very large magnitude of $J_{ij} = 46$ meV, almost double

the thermal energy at room temperature. These calculations together with our experimental results can be understood intuitively as follows: All the Fe atoms in our SL structures are in the half-filled Fe$^{3+}$ state, thus the superexchange interaction between the Fe$_\delta$ spins prefers antiferromagnetic coupling. However, in the case of pairs of (Fe$_i$, Fe$_\delta$) or (Fe$_{As}$, Fe$_\delta$), the atomic distance is much smaller and the direct magnetic coupling, which is FM, becomes dominant. As observed in the STEM and EDX mapping (Fig. 1), the broadening of the Fe distribution in the FeAs layer to the nearest upper and lower layers ($\sigma = 1.5$ ML in Fig. 1d) results in the existence of Fe disorders such as Fe$_i$ or Fe$_{As}$. This disorder-induced FM coupling in the FeAs layers is one of the two origins of the overall ferromagnetism in the FeAs/InAs SLs. The other is the interlayer magnetic coupling between the FeAs layers via the RKKY-like interaction, whose manifestation is the relationship $T_C \propto t_{InAs}^{-3}$ as mentioned above. We note that the disorder-induced intralayer FM coupling does not require itinerant carriers. Meanwhile, the RKKY-like interlayer coupling is rather long-range and requires sufficient itinerancy of carriers. In resistive samples such as sample A4, although the mean free path of carriers is as short as 1.1 nm ($\sim 4$ MLs of InAs) as estimated from the MR results, this mean free path is comparable to the distance $t_{InAs}$ between the FeAs layers ($\sim 5$ MLs of InAs in sample A4). Therefore, it is reasonable to conclude that in all the samples (A1–A4), both the RKKY-like interlayer coupling and disorder-induced intralayer coupling are effective and contribute to establish the overall FM order.

The distribution of Fe in the FeAs/InAs superlattices, particularly the presence of Fe disorders, thus plays a key role in determining the magnetic properties of the overall structures. We conduct X-ray absorption fine structure (XAFS) experiments to characterize the local environment of the Fe atoms, together with our first principles calculations. In Fig. 6a, we show the XAFS spectrum measured at the K-edge of Fe ($\sim 7100$ eV) of the sample A4 (seven FeAs layers, with a distance $t_{InAs}$ of 5 MLs of InAs). In

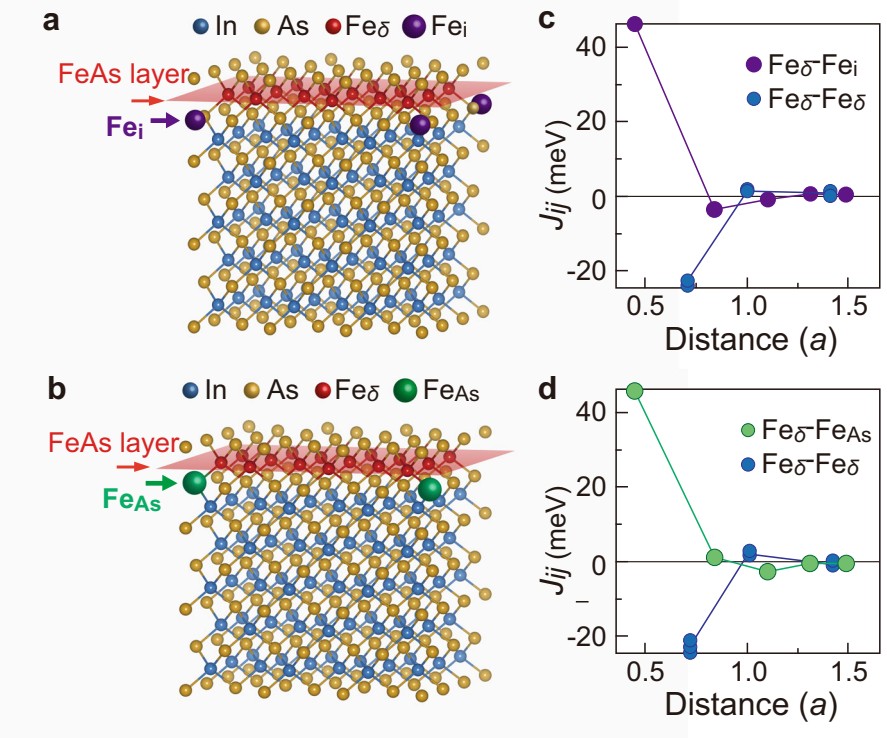

**Fig. 5 First principles calculations with the KKRnano code of Fe–Fe magnetic interactions in FeAs/InAs SLs with Fe disorders. a, b** FeAs/InAs SL structure ($t_{InAs}$ = 5 MLs) with disordered positions of Fe atoms at the octahedral interstitial site (Fe$_i$) and As site (Fe$_{As}$), respectively. **c, d** Magnetic exchange interaction energy $J_{ij}$ between the Fe spins as a function of distance. Here the distance is expressed with a unit of $a$ (= 0.307 nm), which is half of the lattice constant of InAs (in our calculations, we assume that the InAs layers have the same lattice constant as that of the AlSb buffer due to epitaxial strain effect). Positive (negative) values of $J_{ij}$ represent ferromagnetic (antiferromagnetic) coupling between a pair of Fe spins.

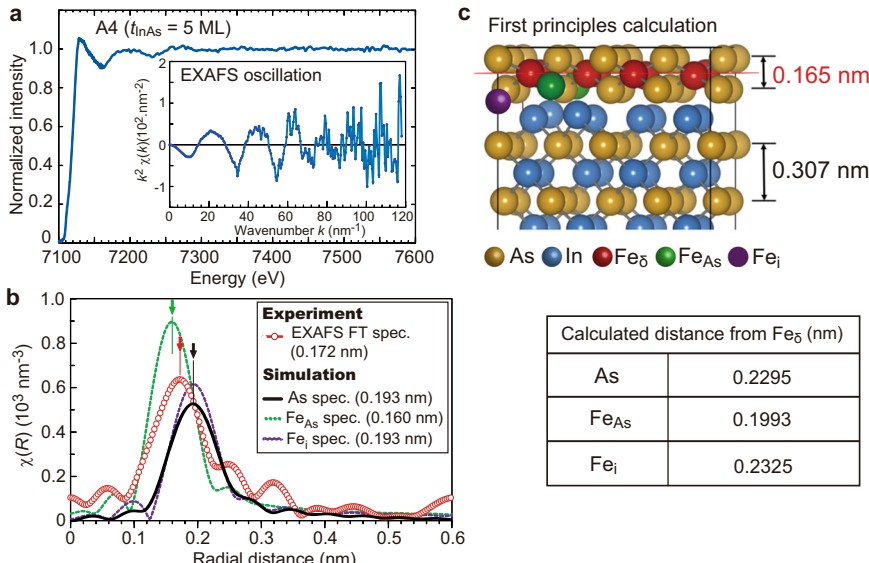

| Calculated distance from Fe$_\delta$ (nm) | |
|---|---|
| As | 0.2295 |
| Fe$_{As}$ | 0.1993 |
| Fe$_i$ | 0.2325 |

**Fig. 6 Local structure around Fe atoms. a** X-ray absorption fine structure (XAFS) spectrum at the Fe K-edge, measured in sample A4 ($t_{InAs}$ = 5 MLs). Inset shows the extended X-ray absorption fine structure (EXAFS) oscillation component extracted from the XAFS spectrum. Here, the EXAFS oscillation is weighted by wavenumber $k^2$. **b** The red curve with white circles shows the Fourier transformed (FT) spectrum of the experimental $k^2$-weighted EXAFS oscillation shown in (**a**) as a function of atomic distance. Black curve is the spectrum simulated from the nearest As atoms in the FeAs monolayer, whose lattice in the $z$ direction shrinks as mentioned in (**c**). Green and purple dotted curves are the spectra simulated from the two Fe disorder positions, As-antisite Fe (Fe$_{As}$) and octahedral interstitial Fe (Fe$_i$), respectively, using the atomic distances obtained by our first principles calculation. The peak position of each curve is pointed by an arrow and provided in the legend. **c** First principles calculation with the VASP code shows that the thickness in the $z$ direction of the FeAs layer shrinks to 0.165 nm, which is only 54% of that (0.307 nm in our calculations) in the InAs layers. Atomic distances from an Fe atom in the lattice site (Fe$_\delta$) to the nearest As, Fe$_{As}$, and Fe$_i$ atoms are also shown in the table.

the inset is the extended X-ray absorption fine structure (EXAFS) oscillation component, whose Fourier transformed spectrum is shown in Fig. 6b (red curve with white circles). There is a large peak at 0.172 nm (pointed by a red arrow), which is much smaller than the distance between the nearest-neighbor atoms (~0.262 nm) in a zinc-blende structure of the host InAs (lattice constant $a = 0.606$ nm). These results imply that there are atoms that reside in a closer vicinity, within the distance to the nearest-neighbor atoms, of the substitutional $Fe_\delta$ atoms in the FeAs layer. We perform a structural optimization using the Vienna ab initio simulation package (VASP) code (see "Methods") and find that the thickness in the $z$ direction of one FeAs monolayer is 0.165 nm, which is only 54% of that in the InAs layer (0.307 nm in our calculations), in order to accommodate the lattice mismatch (Fig. 6c). In this relaxed structure, the distances $d$ from one $Fe_\delta$ atom to the nearest As atom and the two nearest defects, the As-antisite position ($Fe_{As}$) and the octahedral interstitial defect ($Fe_i$), are 0.2295 nm, 0.1993 nm, and 0.2325 nm, respectively, as shown in the table in Fig. 6c. Note that $Fe_{As}$ and $Fe_i$ have strong FM interactions (~46 meV) with the nearest $Fe_\delta$, as explained earlier. Interestingly, the distance $d$ from a $Fe_\delta$ atom to the nearest As-antisite $Fe_{As}$ (0.1993 nm) is shorter than the Fe–As distance (0.2295 nm) because the nearest $Fe_{As}$ and $Fe_\delta$ atoms attract each other, possibly due to their strong FM coupling (see Supplementary Fig. S2). On the other hand, $d$ from a $Fe_\delta$ atom to its second-nearest (the In atoms in the next layer) and third-nearest lattice sites (the next $Fe_\delta$ atoms) are 0.382 nm and 0.428 nm, respectively, which are too far way and not obvious in the experimental curve shown in Fig. 6b. Therefore, the effects of these second- and third-nearest lattice sites are negligible. Using the calculated results, we simulate EXAFS Fourier-transformed spectra coming from the nearest As atoms in the FeAs layer (black curve), and those from the point defects $Fe_{As}$ (green dotted curve) and $Fe_i$ (purple dotted curve), as shown in Fig. 6b. In the ideal case where all the Fe atoms reside in the substitutional positions $Fe_\delta$ of the FeAs layer, the simulated spectrum (black curve) shows a peak at 0.193 nm (pointed by a black arrow), which is still larger than the peak of the experimental spectrum (0.172 nm, pointed by a red arrow). On the other hand, the simulated spectra from $Fe_{As}$ and $Fe_i$ show peaks at 0.160 (pointed by a green arrow) and 0.193 nm, slightly below and above that of the experimental spectrum (0.172 nm). These indicate that only co-existence of different Fe point defects, particularly the antisite $Fe_{As}$ around the $Fe_\delta$ atoms, can explain the EXAFS results. By combining the EXAFS data with the first principles calculations, we conclude that the $Fe_{As}$ and $Fe_i$ point defects are likely responsible for the observed ferromagnetism in the FeAs/InAs.

In conclusion, using the LT-MBE technique, we have successfully grown zinc-blende type FeAs ultrathin layers where the Fe distribution is confined in a thickness of 2–3 MLs. By embedding these FeAs quasi-2D layers in an InAs matrix, we have grown FeAs/InAs SL structures, and revealed several unique features that are promising for spintronic applications. In these FM SL structures, when decreasing the thickness of the InAs spacer ($t_{InAs}$), we observed a drastic enhancement of $T_C$ (proportional to $t_{InAs}^{-3}$), large MR (up to 500% at 2 K in the sample with $t_{InAs} = 5$ MLs), and the very high magnetic moment value of ~5 $\mu_B$ per Fe atom. The very large MR ratios were satisfactorily explained by the theory of GMR in FM/NM multilayers, which suggests that the FeAs layers play an important role of FM layers with a very high spin polarization. The MR characteristics can be modulated by applying a gate voltage. Moreover, the strong enhancement of $T_C$ induced by the Fe disorders and RKKY interaction in these FeAs/InAs SLs paves the way for realizing functional magnetic structures for practical spintronic

applications at room temperature, which utilises state-of-the-art techniques for controlling the atomic distribution in nanostructures.

## Methods

**Calibration of the fluxes of Fe and In**. The In flux was calibrated by monitoring the oscillation in RHEED intensity during the MBE growth of InAs on an InAs (001) substrate. The oscillation period corresponds to the time for growing 1 ML of InAs, which enables us to estimate exactly the growth rate of InAs and the flux of In at various temperatures. In the growth of sample A1–A4, the growth rate of InAs MLs was fixed to 500 nm/h. In our MBE growth, we use a valved cracking cell for As. However, we maintain a low temperature of 600 °C in the cracking part, which means that As is evaporated as $As_4$. We calibrate the fluxes of $As_4$ and In before the low-temperature growth using a beam monitor placed at the sample position. The beam-equivalent-pressure of In is $5 \times 10^{-5}$ Pa, while that of $As_4$ is $2 \times 10^{-4}$ Pa. The Fe flux was calibrated by measuring the Fe concentration in Fe doped GaAs thin films using secondary ion mass spectroscopy (SIMS) calibrated with Rutherford back scattering measurements. In the growth of sample A1–A4, the growth rate of FeAs layers was fixed to 250 nm/h, corresponding to 4 s per 1 ML.

**Preparation of the EDLT device**. The A2 sample was patterned into a $50 \times 200\ \mu m^2$ Hall bar using standard photolithography and ion milling. A side-gate electrode (G) and several electrodes (including the source S and drain D) for transport measurements were formed via electron-beam evaporation and lift-off of an Au (50 nm)/Cr (5 nm) film. The side-gate pad (G) and the FeAs/InAs SL channel were covered with electrolyte (DEME-TFSI) to form the field-effect transistor structure. Other regions of the device were separated from the electrolyte by an insulating resist (OMR-100). As illustrated in Fig. 4a, when a positive $V_G$ is applied, ions in the electrolyte accumulate at the surface of the semiconductor channel and form an electric double-layer capacitor, which works as a nano-scale capacitor. Therefore using $V_G$ we can control the electron concentration in the FeAs/InAs SL structure.

**First principles calculations of the FeAs/InAs SLs**. To study the magnetic properties of the system, we use the KKRnano program developed in Forschungszentrum Jülich on the basis of the all-electron full-potential screened Korringa–Kohn–Rostoker Green's function method[25,26] based on the density functional theory[27,28]. The KKRnano program can calculate very large systems with the order-N method in which the numerical effort is scaled linearly with the number of atoms in the supercell[29–31]. The Vosko–Wilk–Nussair approach to the local density approximation (LDA) was employed for the exchange-correlation functional[32,33]. Relativistic effects were included by means of scalar relativistic approximation[34]. The present calculations were performed with an angular momentum cutoff of $l_{max} = 2$. For the (In,Fe)As, we performed the calculations on a supercell of 432 atomic positions, being occupied by 90 In, 108 As, 18 Fe atoms in the FeAs layer ($Fe_\delta$), and one additionally doped Fe atom ($Fe_i$ or $Fe_{As}$), as shown in Fig. 5. This supercell corresponds to the structure of $3 \times 3 \times 3$ zinc blende unit-cell. To discuss quantitatively the magnetic interactions, we calculate the Heisenberg exchange coupling constant ($J_{ij}$) between the $Fe_\delta$–$Fe_\delta$ atom pairs and that of the $Fe_\delta$–$Fe_i$ atom pairs or $Fe_\delta$ – $Fe_{As}$ atom pairs, on the basis of the Liechtenstein's formula[35]. In the Liechtenstein's formula[35], we consider a perturbation by infinitesimal rotations of magnetic moments, according to the magnetic force theorem[36]. This method has succeeded in quantitatively estimating the magnetic interactions in Fe-based FMS systems.[37,38]

The averaged local magnetic moments in the Voronoi cell calculated by the KKRnano are 3.059 $\mu_B$ and 3.375 $\mu_B$ for the systems in Fig. 5a and b, respectively. These results are smaller than the experimentally observed values (4.7–4.9 $\mu_B$). One reason for the discrepancy is that the present calculations neglect the spin–orbit coupling which leads to finite orbital magnetic moments in lower symmetry systems (see Supplementary Note 1). Another reason is the errors of the LDA. It has been well known that LDA has a difficulty for describing localized $d$ orbitals. The LDA+U or self-interaction correction (SIC) method is often used for the correction of the LDA errors, e.g., the underestimations of exchange splitting and magnetic moments. However, a more important problem is that LDA (even LDA +U and SIC methods) cannot reproduce the finite band gap in the narrow-gap semiconductor InAs. In order to obtain the accurate band gap, we need to employ the hybrid functional or GW method with non-local corrections. With the present computational power, however, it is quite difficult to perform the $J_{ij}$ estimations by the hybrid or GW calculation for the FeAs/InAs superlattices, whose model contains 432 atoms. Such more laborious analysis would be conducted in a future work.

On the other hand, to examine the local atomic structure, we performed the structural optimization using the VASP[39–41], on the basis of the projector augmented wave method[42]. We used a supercell containing 128 atomic positions, being occupied by 56 In, 64 As, 8 Fe atoms in the FeAs layer ($Fe_\delta$), and one additionally doped Fe atom ($Fe_i$ or $Fe_{As}$), as shown in Fig. 6c. This supercell corresponds to the structure of $2 \times 2 \times 4$ zinc blende unit-cell. In the supercell calculations, we use $2 \times 2 \times 1$ Monkhorst-Pack[43] $k$-point meshes and set the plane

wave cutoff energy to 450 eV. The LDA for exchange-correlation functional was employed.[44]

**X-ray absorption fine structure**. XAFS measurements were performed at BL5S1 of Aichi Synchrotron Radiation Center with a Si(111) double-crystal monochromator. The XAFS spectra were obtained in the partial fluorescence yield mode. The X-ray fluorescence signals were detected by an array of seven elements of Si solid state detectors. The monochromator resolution was $E/\Delta E > 7000$. The EXAFS data were analyzed using the Athena and Artemis programs. The $k^2$-weighted EXAFS oscillations $k^2\chi(k)$ were fitted in the region of $k = 2.9$–$8.6$ Å$^{-1}$.

## Data availability

The source data for Figs. 1d, 2, 3, 4, 5c, d, 6a, b in this study are available in the Zenodo database with the following url: https://doi.org/10.5281/zenodo.4783475[45]. Other data that support the findings of this study are available from the corresponding authors upon reasonable request.

## Code availability

The datasets generated during the current study are available from the corresponding authors on reasonable request.

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

## Acknowledgements

This work was partly supported by Grants-in-Aid for Scientific Research (17H04922, 18H05345, 19H05602, 19K21961, 20H05650), CREST program (JPMJCR1777) and PRESTO Program (JPMJPR19LB) of Japan Science and Technology Agency, and Spintronics Research Network of Japan (Spin-RNJ). Y.I. is supported by the A3 Foresight Program. T.F. acknowledges the support from "Building of Consortia for the Development of Human Resources in Science and Technology" and the Supercomputer Center,

the Institute for Solid State Physics, the University of Tokyo. T.F and H.S thank P. H. Dederichs and R. Zeller for many helpful discussions.

## Author contributions

Device fabrication, measurements and data analysis: L.D.A, T.H, Y.N; XAFS data analysis: L.D.A, M.K., First principles calculations: H.S, T.F, H.K-Y., writing and project planning: L.D.A, H.K-Y., Y.I and M.T. All authors extensively discussed the results and the manuscript.

## Competing interests

The authors declare no competing interests.
