## [Peer Review File · Nature Communications]

Reviewers' Comments:

Reviewer #1:

Remarks to the Author:

In “Ferromagnetism and giant magnetoresistance in zinc-blende FeAs monolayers embedded in semiconductor structures” by Anh et al., the authors report ferromagnetism in FeAs/InAs superlattices primarily by combining magnetic, transport and magnetoresistance measurements with first principles calculations. They report – contrary to previous reports – that single layers of Fe-As can exhibit robust magnetism of up to $5 \mu_B$ per Fe, with a highly tunable and large magnetoresistance. They also find that the measured T_C depends on InAs interlayer separation, i.e the out-of-plane distance between Fe-As single layers.

The results here are exciting and are very interesting to the spintronics community and for those broadly interested in Fe-pnictide compounds. However, there are several key weaknesses and inconsistencies with the analysis that need to be addressed before publication:

1. The authors appear to have misunderstood the conclusions from Ref. 12 which they cite throughout the paper. This theory work calculated that yes, indeed, the hypothetical zincblende structure is a half-metallic ferromagnetic. However, the calculated ground state magnetic ordering in this paper is anti-ferromagnetic for this structure – therefore the half-metallic FM order is not expected to form. Their analysis should be revised in light of this throughout the paper.

2. The authors claim that to have grown a SL of FeAs in the InAs matrix, which they depict in the Figures. However, I do not see sufficient evidence for this in the STEM. The authors state that they only see evidence of the InAs matrix in their STEM data and so do not seem to have sufficient evidence to make the claim of a zincblende FeAs layer.

3. Related to their previous point, a key assumption in the analysis is the presence of disorder of the Fe in the InAs matrix – again, what experimental evidence is there for this disorder?

4. The Fe doping percentage considered (1-10%) seems far too low to be in the regime of close Fe-Fe neighbor interactions unless there is clustering of the Fe atoms. In addition to this, Ref [17] shows that a positive exchange is expected for a small range where the distance between the Fe atoms is approximately the order of a lattice constant. This appears to be inconsistent with the conclusion of close nearest-neighbor induced ferromagnetism. The manuscript would be much improved if the estimates of the Fe:As concentrations in the magnetic region were explained, in addition to estimates on the Fe-Fe separation, and whether clustering of the Fe is expected.

5. Related to this previous point – the robust FM with high spin suggests that clusters of bulk Fe could be present in the sample – what evidence do the authors have that this is not the case?

6. I find the report of high-spin FM order on the Fe sites with 5 Bohr magnetons per site surprising for several reasons:

- Firstly, if the disorder-induced magnetism that the authors suggest is correct, I would expect the exchange interactions in the semiconducting phase especially to be anisotropic. In particular, there should be different environments on the Fe sites – and hence a mixture of low-, intermediate- and high-spin states resulting for the various crystal field splittings of the different crystal sites occupied by the Fe atoms. This should be addressed by the authors.

- Secondly, it is a well-known feature of Fe-As compounds that measurements of the magnetic moments often have 'missing' magnetic charge density – see for example Manella *J. Cond. Mat. Phys.* Volume 26, Number 47 (2014). Why do the authors expect that their magnetic moments depart so strongly from previous magnetic measurements in such systems? The authors cite reports of 1.8 Bohr magneton/Fe by other experimental groups in Ref [3]. More analysis to this and an explanation should be included.

- Finally, the transport measurements show a crossover between semiconducting and metallic behavior, however this is not adequately discussed with context to the origins of magnetism. This should also inform the analysis of localized -vs- itinerant magnetism and the corresponding explanations of ferromagnetism. A more careful analysis should be included to point out the changes in expected behavior with these changes in electronic structure.

7. The authors observed that the 'even when Fe atoms are highly concentrated in the 2D ultrathin layers, the band structure of the SLs maintains that of the host InAs'. This is highly surprising in light of the other conclusions made in the paper. Firstly, if ferromagnetism is indeed observed in this system, then the electronic structure should be a half-metallic ferromagnetic, as mentioned by the authors. This should have observable changes in the band structure, notably significant Fe-d states near to the Fermi level.

8. A key part of the analysis is the hypothesis of disorder in the Fe-As layer, which is proposed to explain the observed ferromagnetic behavior. However, the theoretical support is lacking some checks to support this hypothesis, namely:

- What is the energy of defect formation both for the octahedral and As site defects? Based on this magnitude, how many would be expected to form at equilibrium? Is there a steric explanation as to why the Fe would form on this particular sites?

- What is the calculated magnetic moment per Fe site? Is it consistent with the measured value of 5 Bohr magneton/Fe? How anisotropic is the magnetic charge distribution when disorder is introduced?

Some minor points:

1. The authors should not refer to their own results as being 'intriguing'.
2. The authors should give more information and discussion on how the exchange-correlation was described in their first-principles calculations, and how robust their results are to this approximation.

Reviewer #2:

Remarks to the Author:

The manuscript titled by "Ferromagnetism and giant magnetoresistance in zinc-blende FeAs monolayers embedded in semiconductor structures" report a comprehensive study of the structures of quasi two-dimensional (2D) layers of tetrahedral Fe-As bonds embedded with a regular interval in a semiconductor InAs matrix. Contrary to the case of Fe-based pnictides, these FeAs/InAs superlattices (SLs) exhibit ferromagnetism, whose Curie temperature (T_C) increases rapidly with decreasing the InAs interval thickness t_{InAs} , and an extremely large magnetoresistance up to 500% that is tunable by a gate voltage. The authors also carried out first principles calculations, which reveal the important role of disordered positions of Fe atoms in the establishment of ferromagnetism in these quasi-2D FeAs-based SLs. The paper is written clearly, the physics is an interesting topic.

There are several critical issues about the paper.

1. The authors did not cite the related idea of "digital alloy" in the earlier publications, such as Chen et al., Appl. Phys. Lett. 81, 511 (2002) and Soo et al., Appl. Phys. Lett. 80, 2654 (2002) etc. It is important to compare the current results with those from Mn-based ferromagnetic semiconductor quasi-2D SL.
2. Regarding the position of Fe, it is critical to carry out some experiments to resolve it, since this is base for explaining the mechanism of ferromagnetism in this SL. I suggest the authors should conduct EXAFS experiment, which may suggests the local structure around the ferromagnetically ordered Fe atoms in InAs host.
3. The mechanism of RKKY in explaining $T_C \sim t^{-3}$ seems to be reasonable, indicating the plane to plane FM interaction are critical to determine T_C . The lowest T_C in single quasi-2D FeAs layer (sample A1) suggests the ferromagnetic exchange is weak inside 2D plane. The results suggest plane to plane exchange is dominating exchange interaction in these SL. The authors should clarify the link between the in-plane and plane-to-plane exchange interaction: how the plane-to-plane exchange interaction boosts up ferromagnetism. It should be important to compare magnetization per layer in all four samples.
4. Is there Anomalous Hall Effect observed in these samples?
5. What is the easy axis in these samples? Out-of-plane or in-plane?
6. The saturation value in MCD did not depend on the total number of FeAs layer. Why?
7. Are hysteresis loops in MR in agreement with hysteresis loops measured in MCD? How about AHE?

Overall, there are so many unresolved issues in the paper, so I cannot recommend the publication of paper in its current form.

Response Letter to Reviewers' Comments

We would like to thank the reviewers for their valuable comments, which helped us improve the quality of our manuscript. In this Response Letter, we reply to all these comments, point-by-point, and describe how we revised the manuscript. Here, the comments of the reviewers are written by blue characters. Also, the revised parts are written by red characters in this response letter and the revised manuscript.

Reviewer 1:

In "Ferromagnetism and giant magnetoresistance in zinc-blende FeAs mono-layers embedded in semiconductor structures" by Anh et al., the authors report ferromagnetism in FeAs/InAs superlattices primarily by combining magnetic, transport and magnetoresistance measurements with first principles calculations. They report, contrary to previous reports, that single layers of Fe-As can exhibit robust magnetism of up to 5 μB per Fe, with a highly tunable and large magnetoresistance. They also found that the measured TC depends on InAs interlayer separation, i.e the out-of-plane distance between Fe-As single layers. The results here are exciting and are very interesting to the spintronics community and for those broadly interested in Fe-pnictide compounds. However, there are several key weaknesses and inconsistencies with the analysis that need to be addressed before publication:

Our response

We thank the reviewer for the encouraging comments. We would like to address all the questions, point-by-point, and explain the revisions we made in the manuscript, as described below.

Comment 1. The authors appear to have misunderstood the conclusions from Ref. 12 which they cite throughout the paper. This theory work calculated that yes, indeed, the hypothetical zinc-blende structure is a half-metallic ferromagnetic. However, the calculated ground state magnetic ordering in this paper is anti-ferromagnetic for this structure, therefore the half-metallic FM order is not expected to form. Their analysis should be revised in light of this throughout the paper.

Our response

We thank the reviewer for the comment. We understand from the work of Griffin and Spaldin in Ref.12 that the magnetic ground state of zinc blende FeAs is antiferromagnetic. Therefore, in the previous manuscript we cited Ref. 12 which suggests that the ground state in FeAs/InAs zinc blende structures should also be antiferromagnetic, but it turned

out to be contradictory to our experimental results. Specifically, we wrote in lines 52-55
of our previous manuscript as follows:

“A pioneering theoretical work by Griffin and Spaldin¹² suggested that in superlattice
(SL) structures of FeAs tetrahedral layers embedded in a zinc-blende semiconductor
matrix, the antiferromagnetic phase should be the magnetic ground state as in the Fe-
based superconductors.”

The Ref. 12 also pointed out that if we assume a *hypothetical ferromagnetic zinc*
*blende* FeAs phase, the density of states is half-metallic. To clearly state this result, we
revised our manuscript by adding the red texts in line 55: “The work also revealed a half-
metallic band structure in the hypothetical **ferromagnetic phase** of zinc blende FeAs.”

We would like to emphasize that we did *not* use the assumption of half-metallicity of
the hypothetical zinc blende FeAs bulk in our analysis. The ferromagnetic ordering in our
FeAs/InAs superlattice structures are observed consistently by using various
characterization methods such as superconducting quantum interference device (SQUID)
magnetometry, magnetic circular dichroism (MCD), and magnetotransport measurements.
From the observed giant magnetoresistance, we (just merely) suggested that the
*ferromagnetic FeAs monolayers* in our superlattices may also have high spin polarization
similar to the calculated results in the hypothetical ferromagnetic zinc blende FeAs bulk.

**Comment 2.** The authors claim that to have grown a SL of FeAs in the InAs matrix,
which they depict in the Figures. However, I do not see sufficient evidence for this in the
STEM. The authors state that they only see evidence of the InAs matrix in their STEM
data and so do not seem to have sufficient evidence to make the claim of a zinc-blende
FeAs layer.

**Our response**

Our conclusion that the FeAs layers are zinc-blende type crystals was stated based not
only on scanning transmission electron microscopy (STEM) data, but also energy
dispersive X-ray spectroscopy (EDX) mapping of Fe atoms of the same area, as shown
in Fig 1 of the main manuscript (shown again in Fig. R1a). The EDX mapping of Fe
atoms (yellow points) helps us identify the position of the FeAs layer in the InAs matrix.
In the STEM image, the FeAs layer corresponds to a nearly identical darker line located
at 10 nm from the surface. In Fig. R1b, we show a magnified STEM lattice image of the
area surrounded by the blue dashed lines. It is clear that the FeAs layer maintains the
same zinc-blende structure as the surrounding InAs matrix. It is noteworthy that in the in-
plane [-110] direction, the FeAs layer seems to periodically broaden with a period of ~3
72 nm. This possibly results from a spinodal decomposition of Fe. The broadening FeAs

areas may include Fe point defects such as the As-antisite Fe_{As} and octahedral interstitial
 Fe_i , which are found to be important to stabilize the ferromagnetism from our first
 principles calculation in the manuscript.

 **Fig R1. (a) Scanning transmission electron microscopy (STEM) lattice image (left**
 **panel) and energy dispersive X-ray spectroscopy (EDX) mapping of Fe atoms**
 **(yellow points, right panel) in the sample of one FeAs layer embedded in an InAs**
 **matrix. From the EDX mapping results, the FeAs layer position can be identified.**
 **(b) Magnified STEM lattice image of the FeAs layer embedded in the InAs matrix.**
 **The whole area, including the FeAs layer, preserves the zinc blende crystal structure.**

➤ **Corresponding revised parts in the manuscript:**

- ♦ We added Fig. R1 as Supplementary Fig. S1 in Supplementary Information.
- ♦ We revised and added a comment in line 127, page 5 of the main manuscript:

(Before) EDX mapping of the Fe atoms show a similar but clearer image of the Fe distribution (right panel of Fig. 1c)

(After) “EDX mapping results of the Fe atoms also confirm the Fe distribution in a quasi-monolayer located at 10 nm from the surface (right panel of Fig. 1c)”

**Comment 3.** Related to their previous point, a key assumption in the analysis is the
presence of disorder of the Fe in the InAs matrix, again, what experimental evidence is
there for this disorder?

**Our response**

The distribution of Fe in the FeAs/InAs superlattices, particularly the presence
of Fe disorders (point defects as described below), plays a key role in determining the
magnetic properties of the overall structures. Therefore, they deserve further detailed
study, both experimentally and theoretically, as suggested by the two reviewers.
Following their suggestions, we have conducted X-ray absorption fine structure (XAFS)
measurements to characterize the local environment of the Fe atoms, together with our
first principles calculations.

In Fig. R2a, we show the XAFS spectrum measured at the K-edge of Fe (~7100
100 eV) of the sample A4 (7 FeAs layers, with a distance t_{InAs} of 5 monolayers (ML) of InAs).
The inset shows the extended X-ray absorption fine structure (EXAFS) oscillation
component, whose Fourier transformed (FT) spectrum is shown in Fig. R2b (red curve
with white circles). The FT spectrum has a large peak at 0.172 nm (pointed by a red
arrow), which is much smaller than the distance between the nearest-neighbor atoms
(~0.262 nm) in a zinc-blende structure of the host InAs (lattice constant $a = 0.606$ nm).
These results imply that there are atoms that reside in a closer vicinity, within the distance
to the nearest-neighbor atoms, of the substitutional Fe_δ atoms in the FeAs layer.

To explain the EXAFS results, we further examine the local atomic structure
around the FeAs layer using first principles calculations. The structural optimization was
performed using the Vienna *ab initio* simulation package (VASP) code, whose details are
given in the Methods of the revised main manuscript. As shown in Fig. R2c, our results
indicate that the thickness in the z direction of one FeAs monolayer is 0.165 nm, which
is only 54 % of that in the InAs layer (0.308 nm), in order to accommodate the lattice
mismatch. In this relaxed structure, the distances d from one Fe_δ atom to the nearest As
atom and two of the nearest defects, the As-antisite position (Fe_{As}) and the octahedral
interstitial defect (Fe_i), are 0.2295 nm, 0.1993 nm, and 0.2325 nm, respectively, as shown
in the table on Fig. R2c. Note that Fe_{As} and Fe_i have strong ferromagnetic interactions
(~46 meV) with the nearest Fe_δ , as explained in the previous manuscript. Interestingly,

the distance d between an Fe_δ atom and the nearest As-antisite Fe_{As} (0.1993 nm) is shorter
than the Fe-As distance (0.2295 nm) in the same FeAs plane. As shown in Fig. R3, the
nearest Fe_{As} and Fe_δ atoms attract each other, possibly due to their strong ferromagnetic
coupling, resulting in the smaller d . On the other hand, d from a Fe_δ atom to its second-
nearest lattice sites (the In atoms in the next layer) and third-nearest lattice sites (the next
Fe_δ atoms) are 0.382 nm and 0.428 nm, respectively, which are too far way and not
obvious in the experimental curve shown in Fig. R2b. Therefore, the effects of these
second- and third-nearest lattice sites are negligible.

Using the atomic distances from Fe_δ obtained by the first principles calculations,
we simulate EXAFS FT spectra coming from the nearest As atoms in the FeAs layer
(black curve), and those from the point defects Fe_{As} (green dotted curve) and Fe_i (purple
dotted curve), as shown in Fig. R2b. In the ideal case where all the Fe atoms reside in the
substitutional positions (Fe_δ) of the FeAs layer, the simulated spectrum (black curve)
shows a peak at 0.193 nm (pointed by a black arrow), which is still larger than the peak
of the experimental spectrum (0.172 nm, pointed by a red arrow). On the other hand, the
simulated spectra from Fe_{As} and Fe_i show peaks at 0.160 nm (pointed by a green arrow)
and 0.193 nm, respectively, which are slightly below and above that of the experimental
spectrum (0.172 nm). These results indicate that only co-existence of different Fe point
defects, particularly the As-antisite Fe_{As} around the Fe_δ atoms, can explain the EXAFS
results, and thus confirms our claim in the previous main manuscript (*). By combining
the EXAFS data with the first principles calculations, we conclude that the Fe_{As} and Fe_i
point defects are likely responsible for the observed ferromagnetism in the FeAs/InAs
superlattices.

*(*)Since the Fe defects are expected to distribute randomly around the FeAs layer, it is*
*difficult to perfectly fit the simulation to the experimental spectrum.*

➤ **Corresponding revised parts in the manuscript:**

- ♦ We added Fig. R2 as Fig. 6 in the main manuscript.
- ♦ We added the description of EXAFS measurements in Methods in the main
manuscript.
- ♦ We added Fig. R3 as Supplementary Fig. S3 in the main manuscript.
- ♦ We added the above-mentioned discussion about the local structure around Fe
atoms in subsection Discussion (line 305, page13) in the main manuscript.
- ♦ We added the description of the calculations based on the VASP code and related
references in subsection Methods (line 404, page 17) in the main manuscript.

**Fig. R2. (a)** X-ray absorption fine structure (XAFS) spectrum at the Fe K-edge,
 measured in sample A4 ($t_{\text{InAs}} = 5 \text{ ML}$). Inset shows the extended X-ray absorption
 fine structure (EXAFS) oscillation component extracted from the XAFS spectrum.
 Here, the EXAFS oscillation is weighted by wavenumber k^2 . (b) The red curve with
 white circles shows the Fourier transformed (FT) spectrum of the experimental k^2 -
 weighted EXAFS oscillation shown in (a) as a function of atomic distance. Black
 curve is the spectrum simulated from the nearest As atoms in the FeAs monolayer,
 whose lattice in the z direction shrinks as mentioned in (c). Green and purple dotted
 curves are the spectra simulated from the two Fe disorder positions, As-antisite Fe
 (Fe_{As}) and octahedral interstitial Fe (Fe_i), respectively, using the atomic distances
 obtained by our first principles calculation. The peak position of each curve is
 pointed by an arrow and provided in the legend. (c) First principles calculation
 shows that the thickness in the z direction of the FeAs layer shrinks to 0.165 nm,
 which is only 54 % of that (0.308 nm) in the InAs layers. Atomic distances from an
 Fe atom in the lattice site (Fe_δ) to the nearest As, Fe_{As} , and Fe_i atoms are also shown
 in the table.

**Fig. R3.** Side and top views of atomic positions in the case there is one As-antisite Fe
 (**Fe_{As}**, green ball) point defect, obtained by our first principles calculations. The
 nearest Fe_{As} and Fe δ atoms attract each other, possibly due to their strong
 ferromagnetic coupling, resulting in the smaller atomic distance (0.1993 nm)
 than the Fe-As bond length (0.2295 nm).

**Comment 4.** The Fe doping percentage considered (1-10%) seems far too low to be in
 the regime of close Fe-Fe neighbor interactions unless there is clustering of the Fe atoms.
 In addition to this, Ref [17] shows that a positive exchange is expected for a small range
 where the distance between the Fe atoms is approximately the order of a lattice constant.
 This appears to be inconsistent with the conclusion of close nearest-neighbor induced
 ferromagnetism. The manuscript would be much improved if the estimates of the Fe:As
 concentrations in the magnetic region were explained, in addition to estimates on the Fe-
 Fe separation, and whether clustering of the Fe is expected.

**Our response**

In the FeAs/InAs superlattice structures studied here, the Fe atoms are doped
 digitally in only one monolayer (the FeAs layer), which is sandwiched between InAs
 layers. The successful growth of single-crystal zinc-blende type FeAs layers without any
 clustering of Fe has been clearly proved by the STEM and EDX mapping results shown

in Fig. 1 of the main manuscript and Fig. R1 in this response letter. Therefore, although
the average Fe doping percentage is only 1-10%, the local Fe concentration in the FeAs
layers is expected to be close to 100%. In this limit, the Fe-Fe pairs in the vicinity of the
FeAs layers are in next nearest neighbor sites. As indicated by our first principles
calculations in the main manuscript, the interactions between the Fe atoms in the lattice
site (Fe_δ) and the Fe atoms in some disorder positions (Fe_{As} , Fe_i) are strongly
ferromagnetic. We consider these ferromagnetic interactions play a crucial role in
establishing the overall ferromagnetic ground state.

**Comment 5.** Related to this previous point, the robust FM with high spin suggests that
clusters of bulk Fe could be present in the sample. what evidence do the authors have that
this is not the case?

**Our response**

We consider that the magnetic moment of 5 Bohr magneton (μ_B) per Fe atom
clearly excludes the presence of any cluster of bulk Fe. As shown in Fig. R4, the bulk bcc
Fe is known to have a magnetic moment of only $\sim 2 \mu_B/\text{Fe}$ atom. Although the magnetic
moment per Fe atom in smaller Fe clusters slightly increases with decreasing the cluster
size, it is far below the value of $4.7 - 4.9 \mu_B/\text{Fe}$ observed in our FeAs/InAs superlattices.
As can be seen in Fig. R4, the value of $5 \mu_B/\text{Fe}$ just can be reached in the limit of a single
Fe atom, which suggests that there are no Fe clusters in our samples. This conclusion is
also consistent with our microscopic structure characterizations using STEM and EDX
mapping of the FeAs/InAs structures, as described in our response to comment 2.

➤ **Corresponding revised parts in the manuscript:**

- ♦ We added the following comment in line 186, page 8 of the main manuscript: “This
high value of magnetic moment also excludes the possibility of Fe clusters, which
usually have a magnetic moment of only $2.2 \mu_B/\text{Fe}$ atom (see Supplementary Note
1 and Supplementary Fig. S2).”
- ♦ We added Fig. R4 as Supplementary Fig. S2 in Supplementary Information.

**Fig. R4 Magnetic moment per atom in Fe clusters depending on the cluster size**
 **(number of Fe atoms in the cluster) [adapted from J. Meyer et al., J. Chem. Phys.**
 **143, 104302 (2015)].**

**Comment 6a.** I found the report of high-spin FM order on the Fe sites with 5 Bohr
 magnetons per site surprising for several reasons:

- Firstly, if the disorder-induced magnetism that the authors suggest is correct, I would
 expect the exchange interactions in the semiconducting phase especially to be anisotropic.
 In particular, there should be different environments on the Fe sites, and hence a mixture
 of low-, intermediate- and high-spin states resulting for the various crystal field splitting
 of the different crystal sites occupied by the Fe atoms. This should be addressed by the
 authors.

- Secondly, it is a well-known feature of Fe-As compounds that measurements of the
 magnetic moments often have 'missing' magnetic charge density, see for example Manella
 236 J. Cond. Mat. Phys. Volume 26, Number 47 (2014). Why do the authors expect that their
 magnetic moments depart so strongly from previous magnetic measurements in such
 systems? The authors cite reports of 1.8 Bohr magneton/Fe by other experimental groups
 in Ref [3]. More analysis to this and an explanation should be included.

**Our response**

We appreciate these very insightful comments of the reviewer. The observation
of a large magnetic moment close to $5 \mu_B/\text{Fe}$ in our FeAs/InAs superlattices is indeed
surprising, considering previous values reported in other Fe-As compounds. As can be
seen from Fig. R4, the magnetic moment of an isolated Fe atom can reach $6 \mu_B/\text{Fe}$,
consisting of a spin moment of $4 \mu_B/\text{Fe}$ and an orbital moment of $2 \mu_B/\text{Fe}$. Therefore, if
we take into account the orbital moment contribution, a value of $5 \mu_B/\text{Fe}$ is not yet the
possible maximum value as stated in the previous manuscript. We consider that this 5
μ_B/Fe is only an average value of the sum of *the spin moments and orbital moments of Fe*
*atoms* in the lattice sites and defect sites of the FeAs layers, and *the magnetic moments of*
*electron carriers* which are also spin-polarized due to interactions with Fe spins.

In particular, the orbital moment of a Fe atom in a compound depends strongly
on the size and geometry of the compound. For example, in bulk Fe the orbital moment
is frozen out to nearly zero. However, when the cluster size is scaled down, the orbital
moment of the Fe atoms at the surface/interface is known to increase due to the lower
symmetry [see discussion on the scaling laws of spin and orbital moments in Meyer et al.
256 J. Chem. Phys. 143, 104302 (2015) and the references therein]. In our FeAs/InAs
superlattices, the Fe atoms are distributed only in ~ 1 monolayer, neighboring to the InAs
layers in both top and bottom interfaces. This *two-dimensional* geometry maximizes the
number of Fe atoms at the interfaces, which is likely to induce *a large orbital moment*
*component* in the total magnetic moment. Furthermore, as indicated by the first principles
calculations described in the previous manuscript and the XAFS measurements described
in our response to comment 3 and Fig. R2, the most likely positions of Fe in the
FeAs/InAs superlattice structures are the lattice site in the FeAs monolayer (Fe_δ), the As-
antisite position (Fe_{As}), and the octahedral interstitial defect (Fe_i). The interstitial Fe_i
atoms form no bonds with surrounding atoms, thus are close to the situation of an isolated
Fe. This suggests that the Fe_i atoms might have a high magnetic moment close to $6 \mu_B/\text{Fe}$
and largely contribute to the total magnetic moment. The existence of these Fe defects is
different from other Fe-As compounds usually studied in the context of Fe-based
superconductors.

Regarding the magnetic moments of electron carriers, strong magnetic circular
dichroism (MCD) signals (~ 100 mdeg, as shown in Fig. 2 in the main manuscript)
indicate large spin-splitting in the zinc-blende type band structure. Furthermore, the giant
magnetoresistances ($\sim 100\%$, as shown in Fig. 4 of the main manuscript) observed in these
structures clearly indicate a strongly spin-polarized density of states. However, since the
electron density in the system is only of the order of $10^{18}\sim 10^{19} \text{ cm}^{-3}$, which is two orders
of magnitude smaller than the average Fe density ($10^{20}\sim 10^{21} \text{ cm}^{-3}$), we think that the

magnetic moments contributed from electron carriers are negligible.

➤ **Corresponding revised parts in the manuscript:**

- ♦ We added Fig. R4 as Supplementary Fig. S2 in the Supplementary Information.
♦ We added the discussion about the high magnetic moment per Fe as
Supplementary Note 1 in the Supplementary Information.
♦ We revised the expression in line 83, page 4 as follows:

(Before) we find that these FeAs/InAs SLs exhibit strong ferromagnetism whose T_C increases rapidly with decreasing the InAs interlayer thickness t_{InAs} ($T_C \propto t_{\text{InAs}}^{-3}$), with a **perfect** magnetic moment per Fe atom (5 Bohr magneton μ_B)

(After) ...we find that these FeAs/InAs SLs exhibit strong ferromagnetism whose T_C increases rapidly with decreasing the InAs interlayer thickness t_{InAs} ($T_C \propto t_{\text{InAs}}^{-3}$), with a **very high** magnetic moment per Fe atom (**4.7 – 4.9** μ_B , where μ_B is Bohr magneton).

- ♦ We added in the main manuscript the following comment in line 183 page 8:
“**These impressive results might be induced by the 2D distribution of the Fe atoms,**
**which are all neighboring to InAs at both the top and bottom interfaces. ... to as**
**high as that of an isolated Fe atom ($2\mu_B/\text{Fe}$).”**
- ♦ We add Meyer et al. J. Chem. Phys. 143, 104302 (2015) as ref. 23 in the main
manuscript.

**Comment 6b:** Finally, the transport measurements show a crossover between
semiconducting and metallic behavior, however this is not adequately discussed with
context to the origins of magnetism. This should also inform the analysis of localized -
vs- itinerant magnetism and the corresponding explanations of ferromagnetism. A more
careful analysis should be included to point out the changes in expected behavior with
these changes in electronic structure.

**Our response**

As can be seen in Fig. 3 of the main manuscript, the resistances of samples A1
to A4 increase with decreasing the distance t_{InAs} between the FeAs layers. A crossover
between semiconducting and metallic behavior is observed between sample A2 and
sample A3, as noted by the reviewer.

First of all, we would like to note that the increase of resistivity occurs
simultaneously with an increase of the magnetoresistance (MR) in these samples. This
indicates that there is increasingly strong spin-dependent scattering when going from
sample A1 to sample A4, where the distance between the FeAs layers (t_{InAs}) is decreased
from 20 InAs monolayers to 5 InAs monolayers. This strong spin-dependent scattering

causes localization of the electron carriers and is suppressed by applying a magnetic field
to align all the spins. This has been discussed in our previous manuscript (line 216-218
in page 9 of the previous manuscript).

Regarding the origin of the ferromagnetism in our FeAs/InAs superlattice
structures, there are two main mechanisms: (1) The ferromagnetic order in the FeAs
planes is mainly induced by direct exchange between Fe-Fe pairs in the close neighboring
sites, as indicated by our first principles calculations, and (2) the ferromagnetic coupling
between FeAs layers is induced by RKKY-like interlayer coupling, as evident in the t_{InAs} ⁻³
³-dependence of the Curie temperature. Mechanism (1) does not require itinerant carriers
and it is effective in both metallic and semiconducting sides. Meanwhile, mechanism (2)
is rather long-range interaction and requires sufficiently itinerant carriers. By fitting our
model to the t_{InAs} -dependence of the MR magnitude (Fig. 3e), we estimated a mean free
path as short as 1.2 nm (~4 monolayer of InAs) for electron carriers [see subsection
“Magnetoresistances (MR) in the FeAs/InAs SL structures” in page 10 of the main
manuscript]. This mean free path is, however, comparable to the distance t_{InAs} between
the FeAs layers in sample A4 (~5 monolayer of InAs). Therefore, even in samples with
small t_{InAs} and strong spin-dependent scattering, the mechanism (2) is still effective and
helps establishing the overall ferromagnetic order.

➤ **Corresponding revised parts in the manuscript:**

We added the discussion on the mechanism for ferromagnetism in FeAs/InAs
superlattices with regard to the carrier itinerancy in line 296, page 13 of the main
manuscript: “We note that the disorder-induced intralayer ferromagnetic coupling does
not require itinerant carriers. ...and disorder-induced intralayer coupling are effective
and contribute to establish the overall ferromagnetic order.”

**Comment 7:** The authors observed that the 'even when Fe atoms are highly concentrated
in the 2D ultrathin layers, the band structure of the SLs maintains that of the host InAs'.
This is highly surprising in light of the other conclusions made in the paper. Firstly, if
ferromagnetism is indeed observed in this system, then the electronic structure should be
a half-metallic ferromagnetic, as mentioned by the authors. This should have observable
changes in the band structure, notably significant Fe-d states near to the Fermi level.

**Our response**

We would like to clarify some points. Firstly, the discussion that “if
ferromagnetism is observed the electronic structure should be a half-metallic
ferromagnetic” was actually referred to the conclusion in ref.12 [Griffin, S. M. & Spaldin,
341 N. A., *Phys. Rev. B* **85**, 155126 (2012)]. In this reference paper, this conclusion was made

in a *bulk* zinc-blende FeAs compound, which is very different from our systems of FeAs
 *monolayers* in an InAs matrix. Secondly, when we mentioned that “the band structure of
 the SLs maintains that of the host InAs”, what we meant is that the band structure of
 FeAs/InAs maintains the *basic* characteristics of the zinc-blende InAs band structure such
 as the direct band gap and most of the band dispersions. However, there are of course a
 large spin splitting induced by ferromagnetism and Fe-related new bands. We revised the
 corresponding parts in the manuscript to avoid any possible misunderstanding.

Fig. R5. (a) MCD measurement principle. The difference in reflectivity of the right and left circular polarized light comes from the large Zeeman splitting of the band structure of magnetic materials. In the right panel, the absorption MCD spectrum (black curve) and reflection MCD spectrum (green curve) of a FMS are illustrated. (b) MCD spectrum of a nonmagnetic InAs sample under a strong magnetic field (5 T). The MCD peaks correspond to band gaps at critical points of the InAs band structure [Adapted from K. Ando et al., JMMM 272-276, 2004 (2004)]. (c) (=Fig. 2a in the main manuscript) MCD spectra of samples A0 [(In,Fe)As], A1-A4 (FeAs/InAs superlattices), respectively.

The detailed spin-related band structure of FeAs/InAs is not yet clear at this stage.
 However, we reached some important conclusions on the band structure of the FeAs/InAs
 superlattices based on the data presented in the manuscript. The first crucial data are the
 magnetic circular dichroism (MCD) results. As illustrated in Fig R5a, MCD measures the
 difference in optical reflectivity between right (R_{σ^+}) and left (R_{σ^-}) circular polarized light,

which is induced by the spin splitting of the band structure due to a magnetic field or
 magnetization. The MCD intensity is expressed by
$$\text{MCD} = \frac{90}{\pi} \frac{(R_{\sigma+} - R_{\sigma-})}{2R} \propto \frac{90}{\pi} \frac{1}{2R} \frac{dR}{dE} \Delta E$$
, where R is the reflectivity, E is the photon
 energy, and ΔE is the spin-splitting energy (Zeeman energy) of the material. Because
 of the dR/dE term, a MCD spectrum shows peaks corresponding to the optical transitions
 at critical point energies of the band structure of the studied material. At the same time,
 the MCD intensity is proportional to ΔE of the band structure, which in turn reflects the
 magnetization M of the material. Therefore, MCD measurements give information of both
 the magnetization and the electronic structure of the material. An illustrative example is
 given in Fig. R5b, which shows a MCD spectrum of a nonmagnetic InAs under a large
 magnetic field of 5 T. The small Zeeman splitting in the band structure of InAs induces
 peaks at optical transition energies E_1 (2.61 eV), $E_1 + \Delta_1$ (2.88 eV), E_0' (4.39 eV), and E_2
 (4.74 eV). This spectrum serves as a fingerprint of zinc-blend InAs. Given these data, one
 can see that the MCD spectra of Fe doped InAs (sample A0) and all the FeAs/InAs
 superlattices in our study (samples A1-A4) resemble very well that of InAs, but with
 much larger intensity (Fig. R5c). This proves that our FeAs/InAs superlattices preserve
 the basic characteristics of the zinc-blende InAs-type band structure except the large spin
 splitting induced by ferromagnetism and impurity-related new bands.

As shown in Fig. R5c (=Fig. 2a in the main manuscript), the MCD spectra of our
 FeAs/InAs superlattices are very similar to that of the Fe-doped ferromagnetic
 semiconductor (FMS) (In,Fe)As (sample A0). Therefore, their band structures should
 share many common features. Regarding the FMS (In,Fe)As, we have plenty of data
 about its band structure. In Fig. R6a we show the band structure of (In,Fe)As, in
 comparison with that of n-type Be-doped InAs (Fig. R6b), obtained by angular-resolved
 photoemission spectroscopy (ARPES). Because (In,Fe)As is an n-type semiconductor,
 we observed the conduction band bottom and the valence bands (light hole and heavy
 hole bands) below the Fermi level (the zero point of the vertical axis). There are indeed
 Fe d -states (Fe $3d \alpha$ -IB and Fe $3d \beta$ -IB) formed right below the conduction band (and the
 Fermi level). These are fully occupied impurity bands, because the Fe are mainly in the
 form of Fe³⁺ with a half-filled configuration. We note that a large spontaneous spin
 splitting energy of 30 ~ 50 meV was indeed observed in the conduction band bottom of
 (In,Fe)As using a tunneling spectroscopy method [see Anh et al., *Nature Communications*
 **7**, 13810 (2016)]. A similar situation may also be present in the FeAs/InAs superlattices.
 However, we emphasize that the detailed spin-related band structure of FeAs/InAs is not
 yet clear at this stage and deserves more investigations in the future.

Fig R6. Band structures of (a) FMS (In,Fe)As and (b) nonmagnetic InAs, obtained by angular resolved photoemission spectroscopy (ARPES). Under the Fermi level position (energy 0) at the Γ point, the conduction band bottom (electron pocket), light hole (LH), and heavy hole (HH) bands are clearly observed. The Fe-related impurity bands (α -IB and β -IB) in (In,Fe)As are observed by measuring at the resonant photon energy of Fe core levels (Data adapted from M. Kobayashi, LDA et al., arXiv:2009.06285, to be published in Phys. Rev. B).

➤ **Corresponding revised parts in the manuscript:**

To avoid possible misunderstanding, we revised the following parts in the main manuscript:

- ◆ Line 233, page 10, about the possible band structure of bulk FeAs:

(Before) Second, it may originate from the half-metallic DOS of the FeAs layers predicted by the first principles calculation.

(After) **Second, it may originate from high spin-polarization in the DOS of the zinc-blende FeAs layers, because a half-metallic DOS was predicted for bulk FeAs by the first principles calculations.**

- ◆ Line 145, page 6 about the interpretation of the MCD results

(Before) This result indicates that even when the Fe atoms are highly concentrated in the 2D ultrathin layers, the band structure of the SLs maintains that of the host InAs.

(After) This result indicates that even when the Fe atoms are highly concentrated in the 2D ultrathin layers, the band structure of the SLs maintains **the basic properties of the host InAs, such as the band gaps and most of the band dispersions.**

**Comment 8a.** A key part of the analysis is the hypothesis of disorder in the Fe-As layer,
which is proposed to explain the observed ferromagnetic behavior. However, the
theoretical support is lacking some checks to support this hypothesis, namely:

- What is the energy of defect formation both for the octahedral and As site defects? Based
on this magnitude, how many would be expected to form at equilibrium? Is there a steric
explanation as to why the Fe would form on this particular sites?

**Our response**

The existence of the Fe disorder is consistent with the experimental result by the
EDX mapping, where the Fe atoms are confined within a thickness of 2 - 3 MLs around
the FeAs layer (see Fig. 1 of the main manuscript and Fig. R1 of this Response Letter),
and by the EXAFS data and analysis (see Fig. R2 in this Response Letter). From the Fe-
Fe atomic distances, the antisite Fe_{As} defect and the octahedral interstitial Fe_i are most
likely to be present and responsible for the ferromagnetic behavior, as described in our
response to comment 3.

We also calculated the formation enthalpies of these Fe defects using the Vienna
*ab initio* simulation package (VASP) code, whose details are given in the Methods of the
revised main manuscript. We found that they are 0.435 eV and 2.280 eV for the octahedral
interstitial Fe_i and the antisite Fe_{As} defect, respectively. The formation enthalpy of the
octahedral interstitial Fe_i is the second lowest, after the tetragonal interstitial Fe in the
next InAs layer (see Table R1 and Fig. R7). However, we would like to note that first-
principles calculations are basically valid only in the thermal equilibrium state. In the
present experiments, the FeAs/InAs superlattices were grown at very low temperature
($\sim 220^\circ\text{C}$) which is a non-equilibrium MBE growth process. Thus, the situation may be
considerably different from that at the thermal equilibrium state. Therefore, the antisite
Fe_{As} defects may still be formed although the calculated formation enthalpy is high. We
note that at this low growth temperature ($\sim 220^\circ\text{C}$), the Fe:As and In:As flux ratios are
kept very close to 1:1, which are much lower than those ($\sim 1:10$ to $1:20$) at an equilibrium
growth process of InAs at much higher temperature ($450 - 500^\circ\text{C}$). This low As flux may
induce the formation of As-antisite defects in our samples.

**Table R1. Atomic distance from the nearest Fe_δ , formation enthalpy, and average**
 **magnetic moment per Fe atom of several Fe-defect types, in the order of their**
 **distance from the substitutional Fe_δ atom in the FeAs layer, calculated by first**
 **principles calculations.**

Position in Fig. R7	Fe-defect type	Distance from Fe_δ (Å)	Formation enthalpy (eV)	Average magnetic moment/Fe (μ_B)
(a)	As-antisite (Fe_{As})	1.993	2.280	3.375
(b)	Tetragonal interstitial site 1 ($\text{Fe}_{\text{i-T1}}$, in the FeAs plane)	2.293	1.101	3.385
(c)	Octahedral interstitial site ($\text{Fe}_{\text{i-O}}$)	2.325	0.435	3.059
(d)	Hexagonal interstitial site ($\text{Fe}_{\text{i-H}}$)	2.379	0.443	-
(e)	Tetragonal interstitial site 2 ($\text{Fe}_{\text{i-T2}}$, in the next InAs plane)	2.860	-0.311	3.394
(f)	In-antisite (Fe_{In})	3.665	1.893	3.416

**Fig. R7. Several possible Fe-defect positions (green balls, pointed by black arrows)**
 **assumed in our first principles calculations, as also summarized in Table R1. The**
 **substitutional Fe_δ atoms in the FeAs layer, In atoms and As atoms are shown by red,**
 **gray, and black balls, respectively.**

➤ **Corresponding revised parts in the manuscript:**

We added the calculation results of the atomic distance, formation enthalpy and average
magnetic moment per Fe of various types of Fe defects in Table R1 and Fig. R7 as
Supplementary Table S1 and Supplementary Fig. S4, respectively, and the discussion on
the calculated formation enthalpy values as Supplementary Note 2 in the Supplementary
Information.

**Comment 8b:** What is the calculated magnetic moment per Fe site? Is it consistent with
the measured value of 5 Bohr magneton/Fe? How anisotropic is the magnetic charge
distribution when disorder is introduced?

**Our response**

According to our first principles calculations using the KKRnano program,
which is explained in subsection Methods of the main manuscript, the average magnetic
moments per Fe site in the FeAs/InAs superlattices with different types of Fe point defects
are about $3.4 \mu_B/\text{Fe}$, as summarized in Table R1. These values are still smaller than that
observed experimentally ($4.7 - 4.9 \mu_B/\text{Fe}$) in our study. In first-principles calculations, it
has been well known that the local density approximation (LDA) often fails to describe
the localized d states in transition metal elements, leading to underestimation of the
exchange splitting energies and local magnetic moments. The LDA with the self-
interaction correction (SIC-LDA) method improves these shortcomings and gives
reasonable results consistent with the experiments, such as in the cases of (Ga,Mn)N,
(Ga,Mn)As, (Zn,TM)O, and (Ge,Fe) [*Phys. Status Solidi* **3**, 4155 (2006), *Physica B* **376**,
647 (2006), *Phys. Rev. B* **96**, 104415 (2017)]. In our large-scale all-electron calculations,
however, it is quite difficult to apply the SIC-LDA method; therefore, we are employing
LDA for the exchange correlation functional. Moreover, we do not consider the spin-orbit
interaction in the calculations, thus the contribution of the orbital magnetic moments to
the total moment is ignored. In the present case, since the system behaves as n-type, the
electrons which occupy the minority spin states generate finite orbital magnetic moments.
The LDA error and the ignoring the spin-orbit interaction (i.e., orbital moment) are
thought to be the cause of the deviation between the experimental and calculated Fe
magnetic moments.

**Fig. R8. (a) Atomic structure where the corresponding spin densities of each Fe**
 **atoms (Fe δ and Fe $_{As}$) are shown as the red isosurfaces at a specific value of 0.016**
 **($\hbar/2a_B^3$). Here, a_B (= 0.53 Angstrom) is the Bohr radius of a hydrogen atom. (b) Top**
 **view of the FeAs layer, where the color codes indicate the distribution in the xy plane**
 **of the spin density along the z axis.**

In Fig. R8, we also show spin density of a FeAs/InAs superlattice that contains
 486 an As-antisite Fe $_{As}$ defect, which is the most likely case as indicated by our EXAFS
 measurements (see our response to Comment 3). The spin density was calculated using
 the Vienna *ab initio* simulation package (VASP) code, whose details are given in the
 Methods of the revised main manuscript. Figure R8a shows the atomic structure where
 the corresponding spin densities of each Fe atoms (Fe δ and Fe $_{As}$) are shown as the red
 isosurfaces at a specific value of 0.016 ($\hbar/2a_B^3$). Here, a_B (= 0.53 Angstrom) is the Bohr
 radius of a hydrogen atom. Fig. R8b shows the top view of the FeAs layer, where the
 color codes (red: positive, blue: negative, light blue: zero) indicate the distribution in the
 xy plane of the spin density along the z axis. We found that the spin densities are rather
 localized around the Fe lattice sites, a situation that is favorable to induce the high
 magnetic moment per Fe (4.7 - 4.9 μ_B) found experimentally. One can see that upon
 introducing the Fe $_{As}$ defect, the local spin density of the nearest Fe δ sites is strongly
 distorted, which reflects their strong ferromagnetic coupling. However, because the Fe $_{As}$
 defects are likely to distribute randomly, the overall spin density should have a weak
 anisotropy. We are planning to publish detailed calculation results in the near future.

➤ **Corresponding revised parts in the manuscript:**

We added the discussion on the discrepancy between the experimental and calculated
 average magnetic moments per Fe in the Supplementary Note 1 of Supplementary
 Information.

**Comment 9:** The authors should not refer to their own results as being 'intriguing'.

**Our response**

We revised the sentence as the following:

(Before) In particular, it is extremely intriguing to investigate the nature of the magnetic ground state in the 3D-2D crossover limit...

(After) In particular, it is **required and important** to investigate the nature of the magnetic ground state in the 3D-2D crossover limit...

**Comment 10:** The authors should give more information and discussion on how the
exchange-correlation was described in their first-principles calculations, and how robust
their results are to this approximation.

**Our response**

In this work, we employed the local density approximation (LDA) for
investigating the electronic structure and magnetic properties. The KKR calculations in
the framework of the LDA have been well established for ferromagnetic semiconductors
[Rev. Mod. Phys. 82, 1633 (2010), Rev. Mod. Phys. 87, 1311(2015)]. In particular, the
KKR+LDA method gives consistent results with the experimental data for InAs based
ferromagnetic semiconductors [Phys. Rev. Lett. 81, 3002 (1998)]. Of course, there are
many methods for beyond LDA, such as LDA+U, pseudo-SIC, and hybrid functional.
These functionals correct and change the electronic structures, especially, for the Fe-d
states [Phys. Rev. B 96, 104415 (2017)]. However, it is quite difficult to combine the
present large-scale all-electron calculations with the above functionals. Therefore, we
think that LDA is enough for the discussion in this manuscript

➤ **Corresponding revised parts in the manuscript:**

♦ In order to avoid reader's confusion, we added the following sentence to the line
390 in page 16 of the main manuscript: "The Vosko-Wilk-Nussair (VWN)
approach to the local density approximation (LDA) was ...The present
calculations were performed with an angular momentum cutoff of $l_{\max} = 2$."

♦ We added the following references in the main manuscript.

32 S. H. Vosko, L. Wilk, and M. Nusair, Can. J. Phys. **58**, 1200 (1980).

33 S. H. Vosko and L. Wilk, Phys. Rev. B **22**, 3812 (1980).

34 D. D. Koelling and B. N. Harmon, J. Phys. C 10, 3107 (1977).

**Reviewer 2**

The manuscript titled by “Ferromagnetism and giant magnetoresistance in zinc-blende
FeAs monolayers embedded in semiconductor structures” report a comprehensive study
of the structures of quasi two-dimensional (2D) layers of tetrahedral Fe-As bonds
embedded with a regular interval in a semiconductor InAs matrix. Contrary to the case of
Fe-based pnictides, these FeAs/InAs superlattices (SLs) exhibit ferromagnetism, whose
Curie temperature (T_C) increases rapidly with decreasing the InAs interval thickness
t_{InAs} , and an extremely large magnetoresistance up to 500% that is tunable by a gate
voltage. The authors also carried out first principles calculations, which reveal the
important role of disordered positions of Fe atoms in the establishment of ferromagnetism
in these quasi-2D FeAs-based SLs. The paper is written clearly, the physics is an
interesting topic.

**Our response**

We thank the reviewer for the encouraging comments. Below we would like to address
all the questions, point-by-point, and explain the revisions we made in the manuscript.

**Comment 11:** The authors did not cite the related idea of “digital alloy” in the earlier
publications, such as Chen et al., *Appl. Phys. Lett.* 81, 511 (2002) and Soo et al., *Appl.*
*Phys. Lett.* 80, 2654 (2002) etc. It is important to compare the current results with those
from Mn-based ferromagnetic semiconductor quasi-2D SL.

**Our response**

We thank the reviewer for informing us of this previous works. We added new
references in the introduction paragraph of the main manuscript (page 3), as follows:
“Furthermore, enhancement of ferromagnetism has also been reported in similar magnetic
“digital alloys” of Mn-doped FMSs^{18,19,20}.”

We added three references in the main manuscript:

18. Chen, X., Na, M., Cheon, M., Wang, S., Luo, H., McCombe, B. D., Liu, X., Sasaki,
Y., Wojtowicz, T., Furdyna, J. K., Potashnik, S. J., & Schiffer, P., *Appl. Phys. Lett.* **81**,
511 (2002).

19. Soo, Y. L., Kioseoglou, G., Kim, S., Chen, X., Luo, H., Kao, Y. H., Sasaki, Y., Liu,
X., & Furdyna, J. K., *Appl. Phys. Lett.* **80**, 2654 (2002).

20. Nazmul, A. M., Sugahara, S. & Tanaka, M., *Phys. Rev. B* **67**, 241308(R) (2003).

**Comment 12:** Regarding the position of Fe, it is critical to carry out some experiments
to resolve it, since this is base for explaining the mechanism of ferromagnetism in this
SL. I suggest the authors should conduct EXAFS experiment, which may suggests the

local structure around the ferromagnetically ordered Fe atoms in InAs host.

**Our response**

The distribution of Fe in the FeAs/InAs superlattices, particularly the presence
of Fe disorders (point defects described below), plays a key role in determining the
magnetic properties of the overall structures. Therefore, they deserve further detailed
study, both experimentally and theoretically, as suggested by the two reviewers.
Following their suggestion, we have conducted X-ray absorption fine structure (XAFS)
experiments to characterize the local environment of the Fe atoms, together with our first
principles calculations.

In Fig. R9a, we show the XAFS spectrum measured at the K-edge of Fe (~ 7100
581 eV) of the sample A4 (7 FeAs layers, with a distance t_{InAs} of 5 monolayers (ML) of InAs).
The inset shows the extended X-ray absorption fine structure (EXAFS) oscillation
component, whose Fourier transformed (FT) spectrum is shown in Fig. R9b (red curve
with white circles). The FT spectrum shows a large peak at 0.172 nm (pointed by a red
arrow), which is much smaller than the distance between the nearest-neighbor atoms
(~ 0.262 nm) in a zinc-blende structure of the host InAs (lattice constant $a = 0.606$ nm).
These results imply that there are atoms that reside in a closer vicinity, within the distance
to the nearest-neighbor atoms, of the substitutional Fe_δ atoms in the FeAs layer.

To explain the EXAFS results, we further examine the local atomic structure
around the FeAs layer using first principles calculations. The structural optimization was
performed using the Vienna *ab initio* simulation package (VASP) code, whose details are
given in the Methods of the revised main manuscript. As shown in Fig. R9c, our results
indicate that the thickness in the z direction of one FeAs monolayer is 0.165 nm, which
is only 54 % of that in the InAs layer (0.308 nm), in order to accommodate the lattice
mismatch. In this relaxed structure, the distances d from one Fe_δ atom to the nearest As
atom and two of the nearest defects, the As-antisite position (Fe_{As}) and the octahedral
interstitial defect (Fe_i), are 0.2295 nm, 0.1993 nm, and 0.2325 nm, respectively, as shown
in the table in Fig. R9c. Note that Fe_{As} and Fe_i have strong ferromagnetic interactions
(~ 46 meV) with the nearest Fe_δ , as explained in the previous manuscript. Interestingly,
the distance d between an Fe_δ atom and the nearest As-antisite Fe_{As} (0.1993 nm) is shorter
than the Fe-As distance (0.2295 nm) in the same FeAs plane. As shown in Fig. R3, the
nearest Fe_{As} and Fe_δ atoms attract each other, possibly due to their strong ferromagnetic
coupling, resulting in the smaller d . On the other hand, d from a Fe_δ atom to its second-
nearest (the In atoms in the next layer) and third-nearest lattice sites (the next Fe_δ atoms)
are 0.382 nm and 0.428 nm, respectively, which are too far way and not obvious in the
experimental curve shown in Fig. R9b. Therefore, the effects of these second- and third-

nearest lattice sites are negligible.

 **Fig. R9.** (a) X-ray absorption fine structure (XAFS) spectrum at the Fe K-edge,
 measured in sample A4 ($t_{\text{InAs}} = 5 \text{ ML}$). Inset shows the extended X-ray absorption
 fine structure (EXAFS) oscillation component extracted from the XAFS spectrum.
 Here, the EXAFS oscillation is weighted by wavenumber k^2 . (b) The red curve with
 white circles shows the Fourier transformed (FT) spectrum of the experimental k^2 -
 weighted EXAFS oscillation shown in (a) as a function of atomic distance. Black
 curve is the spectrum simulated from the nearest As atoms in the FeAs monolayer,
 whose lattice in the z direction shrinks as illustrated in (c). Green and purple dotted
 curves are the spectra simulated from the two Fe disorder positions, As-antisite Fe
 (Fe_{As}) and octahedral interstitial Fe (Fe_i), respectively, using the atomic distances
 obtained by our first principles calculation. The peak position of each curve is
 pointed by an arrow and provided in the legend. (c) First principles calculation
 shows that the thickness in the z direction of the FeAs layer shrinks to 0.165 nm,
 which is only 54 % of that (0.308 nm) in the InAs layers. Corresponding atomic
 distances from an Fe atom in the lattice site (Fe_δ) to the nearest As, Fe_{As} , and Fe_i
 atoms are also shown in the table.

 Using the atomic distances from Fe_δ obtained by the first principles calculations,
 we simulate EXAFS Fourier-transformed spectra coming from the nearest As atoms in
 the FeAs layer (black curve), and those from the point defects Fe_{As} (green dotted curve)
 and Fe_i (purple dotted curve), as shown in Fig. R9b. In the ideal case where all the Fe
 atoms reside in the substitutional positions (Fe_δ) of the FeAs layer, the simulated spectrum

(black curve) shows a peak at 0.193 nm (pointed by a black arrow), which is still larger
than the peak of the experimental spectrum (0.172 nm, pointed by a red arrow). On the
other hand, the simulated spectra from Fe_{As} and Fe_i show peaks at 0.160 nm (pointed by
a green arrow) and 0.193 nm, respectively, which are slightly below and above that of the
experimental spectrum (0.172 nm). These indicate that only co-existence of different Fe
point defects, particularly the antisite Fe_{As} around the Fe_s atoms, can explain the EXAFS
results, and thus confirms our claim in the previous main manuscript (*). By combining
the EXAFS data with the first principles calculations, we conclude that the Fe_{As} and Fe_i
point defects are likely responsible for the observed ferromagnetism in the FeAs/InAs
superlattices.

*(*)Since the Fe defects are expected to distribute randomly (inhomogeneously) around the*
*FeAs layer, it is difficult to perfectly fit the simulation to the experimental spectrum.*

➤ **Corresponding revised parts in the manuscript:**

- ♦ We added Fig. R9 as Fig. 6 in the main manuscript.
- ♦ We added the description of EXAFS measurements in Methods in the main
manuscript.
- ♦ We added the above discussion about the local structure around Fe atoms in
subsection Discussion (line 305, page 13) in the main manuscript.
- ♦ We added the description of the calculations based on the VASP code and related
references in subsection Methods (line 404, page 17) in the main manuscript.

**Comment 13:** The mechanism of RKKY in explaining $T_C \sim t^{-3}$ seems to be reasonable,
indicating the plane to plane FM interaction are critical to determine T_C . The lowest T_C in
single quasi-2D FeAs layer (sample A1) suggests the ferromagnetic exchange is weak
inside 2D plane. The results suggest plane to plane exchange is dominating exchange
interaction in these SL. The authors should clarify the link between the in-plane and
plane-to-plane exchange interaction: how the plane-to-plane exchange interaction boosts
up ferromagnetism. It should be important to compare magnetization per layer in all four
samples.

**Our response**

We appreciate the insightful comment of the reviewer. In our FeAs/InAs
superlattice structures, the global ferromagnetism is established by two mechanisms: (1)
The ferromagnetic order in the FeAs planes is mainly induced by direct exchange between
Fe-Fe pairs in the close neighboring sites, as indicated by our first principles calculations,
and (2) the ferromagnetic coupling between FeAs layers is induced by RKKY-like

interlayer coupling, as evidenced by the t_{InAs}^{-3} -dependence of the Curie temperature.

In particular, mechanism (1) does not require itinerant carriers and it is effective
only in a close vicinity from the FeAs layer. This explains why in sample A1, where one
single quasi-2D FeAs layer is buried in the center of 40 monolayer-thick InAs, the overall
ferromagnetism is weak. When we increase the number of FeAs layers as in samples A2-
A4, the FeAs layers interact via mechanism (2) which is a long-range interlayer
interaction, in addition to the intralayer interactions via mechanism (1). The two
mechanisms are thus closely correlated and both are inevitable to stabilize the
ferromagnetic ground state of the system.

Regarding the comparison of magnetization per layer in the samples, all samples
A2 to A4 possess similar values of $2.42 - 2.49 \times 10^{-5}$ emu/cm² per FeAs layer, measured
by SQUID at 10 K. These values correspond to $4.7 - 4.9 \mu_{\text{B}}$ /Fe atom in all three samples.
Sample A1 has a T_{C} lower than our SQUID system measurable limit and thus its value
cannot be evaluated.

➤ **Corresponding revised parts in the manuscript:**

We added the discussion on the mechanisms of ferromagnetism in FeAs/InAs
superlattices line 296, page 13 of the main manuscript: “We note that the disorder-
induced intralayer ferromagnetic coupling does not require itinerant carriers.
...Therefore, it is reasonable to conclude that in all the samples (A1 - A4), both the
RKKY-like interlayer coupling and disorder-induced intralayer coupling are effective
and contribute to establish the overall ferromagnetic order.”

**Comment 14: Is there Anomalous Hall Effect observed in these samples?**

**Our response**

We observed the anomalous Hall effect in our samples. Figure R10a shows Hall
resistance data measured in sample A3 (5 layers of FeAs) at 3.5 K. The observation is
quite challenging because the anomalous Hall resistances (AHR) in these samples are
small. As can be seen in Fig. R10a, the AHR component is only 5% of the total Hall
resistance, and is almost hidden by the normal Hall resistance component. This feature is
similar to that of n-type FMS (In,Fe)As [see LDA et al., Phys. Rev. B **92**, 161201(R)
(2015)]. This small AHR is expected in n-type ferromagnetic semiconductors because of
the weak spin-orbit interaction in the conduction band (weaker than that in the valence
band). Furthermore, the samples with strong ferromagnetism such as A2, A4 and A4 are
highly resistive, which further hinders the Hall measurements. As a result, the AHR
component is noisy as can be seen in Fig. R10a. Nevertheless, the ferromagnetic
hysteresis loops obtained from AHR, MCD, and SQUID measurements agree very well,

as shown in Fig. R10b. This indicates that the ferromagnetism in the FeAs/InAs samples
is intrinsic and single-phase.

**Fig R10. (a) Hall resistance (purple circles), which includes a normal Hall resistance**
**(dark blue line) and an anomalous Hall resistance (red open circles, right axis),**
**measured in sample A3 at 3.5 K. (b) Normalized hysteresis loops obtained by**
**anomalous Hall effect, MCD, and SQUID measurements in sample A3, which show**
**good agreement.**

➤ **Corresponding revised parts in the manuscript:**

We added Fig. R10 and the abovementioned discussion as Supplementary Fig. S5 and
Supplementary Note 3, respectively, in Supplementary Information.

**Comment 15: What is the easy axis in these samples? Out-of-plane or in-plane?**

**Our response**

Figure R11 shows the magnetization curves measured at 10 K when a magnetic field
is applied perpendicular to the plane and in the plane in the three samples A2, A3, A4.
These normalized hysteresis curves indicate that the easy axis of magnetization in these
samples is in the plane.

➤ **Corresponding revised parts in the manuscript:**

We added Fig. R11 as Supplementary Fig S7 in Supplementary Information.

**Fig. R11. (a),(b),(c) Normalized magnetization curves measured when a magnetic**
 **field is applied perpendicular to the plane (black squares) and in the plane (red**
 **circles) in samples A2, A3, and A4, respectively. All the magnetization curves were**
 **measured at 10 K by SQUID, except the curve with a perpendicular magnetic field**
 **in sample A2 that was measured by MCD.**

Comment 16: The saturation value in MCD did not depend on the total number of FeAs
 layer. Why?

**Our response**

This is an interesting question. As mentioned in our response to comment 13, the
 magnetization per FeAs layer in all the samples are virtually same, with a similar value
 of $2.42 - 2.49 \times 10^{-5}$ emu/cm² per FeAs layer. Therefore, the magnetization in these
 samples should scale up with the number of FeAs layers. Consequently, the MCD
 intensity, which is proportional to the magnetization, should also increase from sample
 A1 to A4. Indeed, the MCD intensity of sample A2 is double that of sample A1. However,
 it stays constant (~100 mdeg) in sample A2, A3 and A4 (see Fig. 2 in the main manuscript
 or Fig. R5 in page 13 of this Response Letter). We think that this is due to the short
 penetration depth of the incoming light in these FeAs/InAs superlattice structures: The
 MCD effectively measures into a depth corresponding to three FeAs/InAs periods. This
 is a possible scenario, because the narrow-gap InAs host strongly absorbs visible light
 (200 – 800 nm). Moreover, the superlattice structure and the mid-gap states introduced
 by the FeAs layers may enhance the light absorption and further limit the penetration
 depth.

➤ **Corresponding revised parts in the manuscript:**

- ♦ We added a comment about the MCD intensity in sample A2, A3, A4 in line 153, page
 7 of the main manuscript, as the follows: “**The MCD intensity in sample A2, A3, A4**
 **stays almost constant, which may be due to a short penetration depth of visible light in**
 **these structures. (See Supplementary Note 4).”**

♦ We added the above-mentioned discussion as the Supplementary Note 4 in
Supplementary Information.

**Comment 17:** Are hysteresis loops in MR in agreement with hysteresis loops measured
in MCD? How about AHE?

**Our response**

As shown in Fig. R12, the hysteresis loop measured with magnetoresistance
(MR) and with MCD in a sample (A4) agree with each other very well (the coercive
forces are ± 188 Oe). The same agreement between the hysteresis loops measured by the
anomalous Hall resistance, MCD, and magnetization was also confirmed as shown in Fig.
R10b (our response to comment 15). These results indicate that the ferromagnetism in the
FeAs/InAs samples is intrinsic and single-phase.

**Fig. R12. Comparison between the coercive forces measured by MCD (blue**
**diamonds) and magnetoresistance (black and red circles) curves at 4 K in sample**
**A4.**

➤ **Corresponding revised parts in the manuscript:**

We added Fig. R12 as Supplementary Fig S6 in Supplementary Information.

Reviewers' Comments:

Reviewer #1:

Remarks to the Author:

The authors have made significant improvements to the paper, and have addressed many of the concerns regarding the previous version.

However, their discussion of first principles accuracy and methodology remain insufficient.

- The authors state that the issue with using LDA for Fe-d states is that it doesn't incorporate spin-orbit interactions. While spin-orbit interactions will of course have an influence on the total moment, the much greater correction comes from the underlocalization of the d manifold from standard LDA calculations. Improvements to such calculations can be made using LDA+U approaches, for example, which have equivalent run times in VASP as standard LDA. Moreover, with the mixture of methodology (KKR -vs- PAW) used in the analysis -- how transferable are the results between the different codes/methods used? Given the misunderstanding of the methodological differences for these calculations, a more thorough analysis of their deviation from the calculated moments (~ 3.5 Bohr magneton in DFT -vs- 5 Bohr magneton experimentally) should be included before publication.

Reviewer #2:

Remarks to the Author:

The authors have extensively revised the paper according to the two reviewers' report.

Importantly, the authors have carried out the XAFS spectrum measurement, and thus have significantly enriched the content of the paper. I am generally satisfied with the authors' response and therefore I would like to recommend the publication of the paper after the authors consider a couple minor issues.

1. What is the As₂:In flux ratio during the LT MBE growth of the SL? Is an As cracker used in the MBE growth?

2. In page 7, line 172, this statement: "In eq. (2), the first term is roughly 4 times larger than the second term for $N = 1, 3, 5, 7$." is not correct for $N = 1$. In fact, $T_c = 0$ for $N = 1$, suggesting the dominant ferromagnetism originates from interlayer RKKY -type exchange.

Second Response Letter to Reviewers' Comments

We would like to thank the reviewers for their valuable comments, which helped us improve the quality of our manuscript. In this Second Response Letter, we reply to all these comments, point-by-point, and describe how we revised the manuscript. Here, the comments of the reviewers are written by blue characters. Also, the revised parts are written by green characters in this response letter and the revised manuscript.

Reviewer 1:

The authors have made significant improvements to the paper, and have addressed many of the concerns regarding the previous version. However, their discussion of first principles accuracy and methodology remain insufficient.

Our response

We thank the reviewer for the encouraging comments. In the following, we would like to address the issues pointed out by the reviewer.

Comment 1a: Moreover, with the mixture of methodology (KKR -vs- PAW) used in the analysis -- how transferable are the results between the different codes/methods used? Given the misunderstanding of the methodological differences for these calculations, a more thorough analysis of their deviation from the calculated moments (~3.5 Bohr magneton in DFT -vs- 5 Bohr magneton experimentally) should be included before publication.

Our response

First, we would like to mention the calculated methods used in our work, which is inquired in comment 1a. As correctly pointed out by the reviewer, in this work we employed two methods for the first-principles analysis, i.e., the Korringa-Kohn-Rostoker (KKR) Green's function method (KKRnano) and the projector augmented wave (PAW) method (Vienna *ab initio* simulation package - VASP), whose details were already described in Methods of the main manuscript. However, the results of these two methods are used separately for different purposes, as explained below.

The KKR Green's function method, combining with the Liechtenstein's formula, allows us to directly calculate the magnetic exchange interaction parameters (J_{ij}) and investigate their dependence on the distance between the Fe atoms in the present quasi-2D FeAs/InAs superlattices. Our KKRnano, which is a full-potential version of the KKR method, is effective even for complex crystal systems. However, it cannot perform

structural optimization because the space is divided into the Voronoi cells. On the other
hand, the VASP code, which is based on the PAW method, allows us to perform structural
optimization and calculate the formation enthalpy. Therefore, we used KKRnano and
VASP *separately* for studying the magnetic properties and structural properties,
respectively, of the FeAs/InAs systems.

We note that it is impossible to directly compare the calculated local magnetic moments
by the KKRnano and those by the VASP codes. This is because the local magnetic
moments deduced in the two methods are calculated differently: They are calculated in
the Voronoi cell for KKRnano but in atomic sphere with a certain radius for VASP,
respectively.

➤ **Corresponding revised parts in the manuscript:**

To avoid any confusion and misunderstanding by readers, we made the following
corrections in the main manuscript:

● We specified which first-principles code (KKRnano or VASP) was used in the
captions of Fig. 5 and Fig. 6 in the main manuscript, and Fig. S3 and Table S1 in the
Supplementary Information.

● In Supplementary Table S1, the calculated results of the two first-principles codes
are mixed. The data in “Distance from Fe_δ” and “Formation enthalpy” columns are
calculated by VASP, while the data in the “Average magnetic moment/Fe” column
are the KKRnano results. To avoid confusion, we deleted the “Average magnetic
moment/Fe” column and moved the information of the calculated magnetic moments
to the subsection “First principles calculations of the FeAs/InAs SLs” in Methods,
from line 410 page 17 in the revised manuscript, mentioned as follows: “**The**
**averaged local magnetic moments in the Voronoi cell calculated by the KKRnano are**
**3.059 μ_B and 3.375 μ_B for the systems in Figs. 5a and 5b, respectively.**”

**Comment 1b:** The authors state that the issue with using LDA for Fe-d states is that it
doesn't incorporate spin-orbit interactions. While spin-orbit interactions will of course
have an influence on the total moment, the much great correction comes from the
underlocalization of the d manifold from standard LDA calculations. Improvements to
such calculations can be made using LDA+U approaches, for example, which have
equivalent run times in VASP as standard LDA.

**Our response**

As pointed out by the reviewer, the magnetic moment of the Fe atoms in the quasi-2D
FeAs/InAs superlattices calculated by the KKRnano with local density approximation

(LDA) is smaller than the experimental values of $4.7 - 4.9 \mu_B$ estimated from the
 magnetization measurements. It is well known that the LDA + U method or variational
 principle pseudo self-interaction correction (VPSIC) method can improve the LDA with
 errors for localized d states. To confirm the suggestion of the reviewer, we estimated the
 magnetic moment per Fe atom in a bulk (In,Fe)As sample doped with 5% Fe by the KKR-
 VPSIC method, which can obtain a correction similar to the LDA+U method, without the
 U parameter [see Phys. Rev. B **84** 195127 (2011), Phys. Rev. B **96** 104415 (2017), Appl.
 Phys. Exp. **12** 063006 (2019)]. Here, we used the KKR-coherent potential approximation
 (CPA) method (AkaiKKR code) with order N^3 computational cost (for more details:
 <http://kkp.issp.u-tokyo.ac.jp>). This is because the VPSIC method is not implemented in
 our order-N KKRnano code used in the present manuscript. Figures R1(a) and 1(b) show
 the calculated density of states of the (In_{0.95}Fe_{0.05})As sample by the LDA and VPSIC
 methods, respectively. Compared with LDA, the VPSIC method makes the Fe- d states
 more localized and moves them toward energetically deeper region. Indeed, the VPSIC
 method leads to a magnetic moment of $3.86 \mu_B$ within the muffin-tin sphere, which is
 slightly larger than the LDA result ($3.64 \mu_B$). However, we were not able to reproduce the
 experimental result even by using the VPSIC method.

**Fig.R1. Density of states of (In_{0.95}Fe_{0.05})As calculated by (a) LDA and (b) VPSIC**
 **methods.**

 One reason for this discrepancy may be that we cannot obtain the finite band gap
 of the host InAs, which is ~ 0.36 eV in experiment, by using either LDA, LDA+U, or
 VPSIC. In order to obtain a more accurate band gap energy value, we need additional
 non-local correction schemes, such as hybrid functional and GW methods. However, with
 the present computational power, it is quite difficult to perform the hybrid or GW
 calculations for the present model of quasi-2D FeAs/InAs superlattices, which contains
 432 atoms. Such more laborious analyses would be targeted in a future work.

Another simple reason for the discrepancy might be that the local magnetic moments
calculated by first-principles calculations depend on the volume of the considered region,
e.g., an atomic sphere with a certain radius. In the case of the KKRnano code, the system
is divided into the Voronoi cells and the local magnetic moment is calculated
within each Voronoi cell, thus the magnetic moment is underestimated.

➤ **Corresponding revised parts in the manuscript:**

To provide readers a more explicit and clear discussion on the discrepancy between the
calculational and experimental local magnetic moments of Fe in our samples, we added
the following sentences in the subsection “First principles calculations of the FeAs/InAs
SLs” in Methods, line 410 page.17 in the revised manuscript:

“The averaged local magnetic moments in the Voronoi cell calculated by the KKRnano
are 3.059 μ_B and 3.375 μ_B for the systems in Figs. 5a and 5b, respectively. These results
are smaller than the experimentally observed values (4.7 – 4.9 μ_B). One reason for the
discrepancy is that the present calculations neglect the spin-orbit coupling which leads to
finite orbital magnetic moments in lower symmetry systems (see Supplementary Note 1).
Another reason is the errors of the LDA. It has been well-known that LDA has a difficulty
for describing localized *d* orbitals. The LDA+U or self-interaction correction (SIC)
method is often used for the correction of LDA errors, e.g., the underestimations of
exchange splitting and magnetic moments. However, a more important problem is that
LDA (even LDA+U and SIC methods) cannot reproduce the finite band gap in the
narrow-gap semiconductor InAs. In order to obtain the accurate band gap, we need to
employ the hybrid functional or GW method with non-local corrections. With the present
computational power, however, it is quite difficult to perform the J_{ij} estimations by the
hybrid or GW calculation for the FeAs/InAs superlattices, whose model contains 432
atoms. Such more laborious analyses would be conducted in a future work.”

Accordingly, we removed the similar discussion at the end of the Supplementary Note 1
of the previous Supplementary Information, to avoid redundancy.

**Reviewer 2:**

The authors have extensively revised the paper according to the two reviewers' report.
Importantly, the authors have carried out the XAFS spectrum measurement, and thus have
significantly enriched the content of the paper. I am generally satisfied with the authors'
responds and therefore I would like to recommend the publication of the paper after the
authors consider a couple minor issues.

**Our response**

We thank the reviewer for the encouraging comments and recommending the
publication of our paper. We would like to address the issues pointed out by the reviewer
as follows.

**Comment 1:** What is the As₂:In flux ration during the LT MBE growth of the SL? Is a
As cracker used in the MBE growth?

**Our response**

In our MBE growth, we use a valved cracking cell for As. However, we maintain a
low temperature of 600°C in the cracking part, which means that As is evaporated as As₄.
We calibrate the fluxes of As₄ and In before the low temperature growth using a beam
monitor placed at the sample position. The beam-equivalent-pressure (BEP) of In is 5×10^{-5}
149 Pa, while that of As₄ is 2×10^{-4} Pa. These BEP values have been optimized for the low-
150 temperature growth of our Fe doped InAs samples.

➤ **Corresponding revised parts in the manuscript:**

We added the description of In and As fluxes in Method, from line 369 – 373, page 16 of
the revised manuscript, as follow:

“In our MBE growth, we use a valved cracking cell for As. However, we maintain a low
temperature of 600°C in the cracking part, which means that As is evaporated as As₄. We
calibrate the fluxes of As₄ and In before the low temperature growth using a beam monitor
placed at the sample position. The beam-equivalent-pressure (BEP) of In is 5×10^{-5} Pa,
while that of As₄ is 2×10^{-4} Pa.”

**Comment 2:** In page 7, line 172, this statement: "In eq. (2), the first term is roughly 4
161 times larger than the second term for N = 1,3,5,7. " is not correct for N = 1. In fact, T_c=0
for N = 1, suggesting the dominant ferromagnetism originates from interlayer RKKY -
type exchange.

**Our response**

We thank the reviewer for the notice. For the sake of accuracy, in the main manuscript
we have revised the sentence as: " In eq. (2), the first term is roughly 1.5, 2.5, 3, and 4

167 times larger than the second term for $N = 1, 3, 5, 7$, respectively. Therefore, considering the
168 relationship $N \propto t_{\text{InAs}}^{-1}$, if we express T_C as $t_{\text{InAs}}^{-\gamma}$ the exponent γ is a value close to 3. As
can be seen in Fig. 2c, the T_C values calculated using eq. (2) with $A = 250$ (open diamonds)
well reproduce the experimental results (pink circles).".

Note that we have also corrected the value of A from 190 (previously) to 250 in the revised
manuscript. This minor correction does not affect any conclusion of our work.

**Other revised parts in the manuscript**

1. We noticed that in the bottom panel of Fig. 2b, which shows the magnetic-field-
dependence of the magnetic circular dichroism (MCD) at various temperatures in
sample A4, the color labels of measurement temperatures (from 5 to 150 K) in the
legend were listed wrongly in an opposite order. Therefore we revised Fig. 2 and
corrected this error.

2. In the caption of Fig. 2, we added a description of the error bars of the experimental
Curie temperatures shown in Fig. 2c, as follows: "The error bars of 5 K are also
plotted, which correspond to the minimum temperature step in our MCD – H
measurements."

Reviewers' Comments:

Reviewer #1:

Remarks to the Author:

The authors have sufficiently addressed my comments and updated their manuscript accordingly.

Reviewer #2:

Remarks to the Author:

The authors have responded all my comments. I would like to recommend the publication of the paper as its current form.

Third Response Letter to Reviewers' Comments

We would like to thank the reviewers for their valuable comments, which helped us improve the quality of our manuscript. In this Response Letter, we reply to all the comments. Here, the comments of the reviewers are written by blue characters.

Reviewer 1:

The authors have sufficiently addressed my comments and updated their manuscript accordingly.

Our response

We appreciate many thoughtful comments raised by the reviewer, which help improving largely the quality and readability of our manuscript.

Reviewer 2:

The authors have responded all my comments. I would like to recommend the publication of the paper as its current form.

Our response

We appreciate many encouraging comments raised by the reviewer. The revisions we made to comply with these comments have improved the quality and readability of our manuscript.

Other minor revisions related to the manuscript content:

We renamed the parameter J_{ij} in lines 167, 169, and equation (1) of the main manuscript to J_{sij} , to distinguish it from the J_{ij} parameter obtained with our first-principle calculations in lines 279, 288, and Fig. 5c,d.